# The Role of Target Update Frequencies in Q-Learning

**Simon Weissmann** [1]  **Tilman Aach** [1]  **Benedikt Wille** [1]  **Sebastian Kassing** [2]  **Leif Döring** [1]

## Abstract

The target network update frequency (TUF) is a central stabilization mechanism in (deep) Q-learning. However, its selection remains poorly understood and is often treated merely as another tunable hyperparameter rather than as a principled design decision. This work provides a theoretical analysis of target fixing in tabular Q-learning through the lens of approximate dynamic programming. We view periodic Q-learning as a nested optimization scheme in which each outer iteration applies an inexact Bellman optimality operator, approximated by a generic inner loop optimizer. Rigorous theory yields a finite-time convergence analysis for the asynchronous sampling setting, specializing to stochastic gradient descent in the inner loop. Our results deliver an explicit characterization of the bias–variance tradeoff induced by the target update period, showing how to optimally set this critical hyperparameter. We prove that constant target update schedules are suboptimal, incurring a logarithmic overhead in sample complexity that is entirely avoidable with adaptive schedules. Our analysis shows that the optimal target update frequency decreases geometrically over the course of the learning process.

## 1. Introduction

Reinforcement learning (RL) is a powerful framework for sequential decision-making, allowing agents to learn optimal behavior through interaction with their environment (Sutton & Barto, 2018). Many successful RL algorithms are rooted in dynamic programming (DP), which takes advantage of the recursive structure of value functions to solve Markov decision processes (MDPs) (Puterman, 2014). How-

ever, classical DP requires explicit knowledge of transition dynamics and becomes intractable in large or continuous state spaces, motivating approximate dynamic programming (ADP) methods that scale to real-world problems. Value-based ADP methods form a core class of RL algorithms, aiming to learn an optimal value function from which an optimal policy can be inferred. Among approximate DP approaches, variants of Q-learning (Watkins & Dayan, 1992) are particularly influential, offering a model-free method to learn the optimal action-value function by iterative application of the Bellman optimality operator $T^*$. Its appeal lies in the simplicity: temporal-difference updates bootstrap value estimates without an explicit environment model. Despite convergence guarantees in the tabular case, Q-learning is unstable with function approximation, especially neural networks, due to the deadly triad of approximation, bootstrapping, and off-policy learning. Deep Q-Networks (DQN) mitigate these issues through experience replay and a target network (Mnih et al., 2015). The target network, a periodically fixed copy of the Q-network, helps to de-correlate targets from current parameters. In tabular form, fixed target Q-learning is defined as

$$Q(s,a) \leftarrow Q(s,a) + \alpha\big(r + \gamma \max_{a'} Q^-(s',a') - Q(s,a)\big),$$

where the fixed target matrix $Q^-$ is periodically overwritten with $Q$, say all $K = 1000$ steps. We will abbreviate the target update frequency $1/K$ by TUF. From an ADP perspective, fixed targets modify how the Bellman optimality operator is approximated. While standard Q-learning minimizes the squared Bellman error using a single stochastic gradient descent (SGD) update, a TUF with parameter $K$ corresponds to performing $K$ consecutive SGD iterations on the same objective (see Section 3 below). Therefore, we will interpret the choice of the TUF as a multilevel design problem and ask if choosing a fixed TUF is optimal.

*Can convergence of Q-learning be improved by systematically* varying *the degree of temporal separation between current estimates and bootstrap targets?*

Our answer is positive. The TUFs should decrease geometrically during training; keeping the cycle length $K$ fixed is suboptimal. Before going into the theory, let us consider two illustrative experiments shown in Figure 1. We consider a delicate tabular GridWorld example and Lunar Lander.

[1]Institute for Mathematics, University of Mannheim [2]Department of Mathematics & Informatics, University of Wuppertal. Correspondence to: Simon Weissmann <simon.weissmann@uni-mannheim.de>, Benedikt Wille <benedikt.wille@uni-mannheim.de>.

*Proceedings of the 43rd International Conference on Machine Learning*, Seoul, South Korea. PMLR 306, 2026. Copyright 2026 by the author(s).

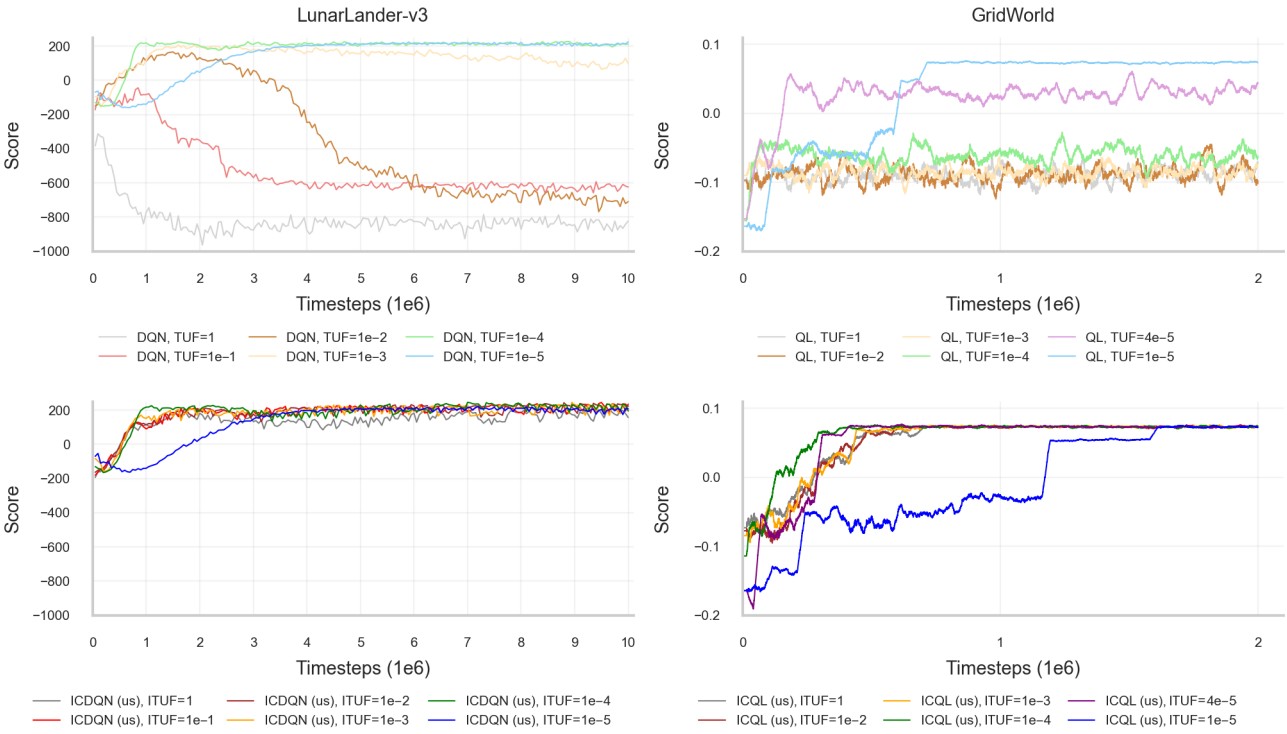

*Figure 1.* Top row: Lunar Lander (Algo: DQN with different constant TUFs using SGD with custom learning rate schedule that linearly decreases from 0.01 to 0.0001 over each target-freezing interval) and a GridWorld from Appendix A.2 (Algo: Q-learning with target freezing and our theory guided learning rate $1/(1 + k/(2|\mathcal{S}||\mathcal{A}|))$ with $|\mathcal{S}||\mathcal{A}| = 52$ over each target-freezing interval). Bottom row: Same environments, same algorithms but decreasing TUF, initialised in fixed TUF from top row. ITUF denotes the initial TUF in our ICQL and ICDQN algorithm.

For the first, we use tabular Q-learning with target fixing (standard Q-learning with constant learning rate is TUF=1). For the second, we use DQN with ADAM replaced by SGD to stay close to our theoretical findings. More details are given in Appendix A.

The plots illustrate that high TUFs sometimes fail entirely, medium frequencies can learn quickly but saturate the learning, and low frequencies learn slowly but successfully. Related work in deep Q-learning has reported similar sensitivity of performance to the choice of the TUF (Jia et al., 2021; Hernandez-Garcia & Sutton, 2019; Sun et al., 2023; Asadi et al., 2023).

From the ADP point of view, the contractive nature of Bellman operators gives a simple intuition that is made rigorous in Section 5: (i) In early Bellman iterations, when current value estimates are far from the optimal fixed point $Q^*$, the contractivity of $T^*$ moves estimates in the correct direction, even for large approximation errors of $T^*$. (ii) As learning progresses and the current estimates $Q$ approach the optimal $Q^*$, the margin for approximation error diminishes. Continued

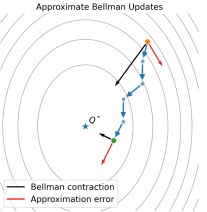

convergence requires increasingly accurate approximations of $T^*$ to avoid domination of approximation errors.

The theoretical analysis of this work shows how this intuition can be transferred to the choice of TUFs. Here is our main theoretical result, which shows that geometrically decreasing TUF provably improves the error dependence in the sample complexity of tabular Q-learning with delayed target updates. A more detailed version is given in Section 5.

**Theorem 1.1** (Informal). *Suppose rewards are bounded and the state-action visitation probabilities are lower bounded by $\xi > 0$. In order to achieve accuracy $\mathbb{E}[\|Q_t - Q^*\|_\infty] < \varepsilon$ with $t$ updates, the following number of samples is needed:*

1. *With optimal constant TUF the sample complexity to reach accuracy $\varepsilon$ is*

$$\mathcal{O}\Big( \frac{\log((1-\gamma)^{-1}\varepsilon^{-1})}{\xi^2(1-\gamma)^4\varepsilon^2 \log(2(1+\gamma)^{-1})} \Big).$$

2. *With geometrically (of order $\gamma^{\frac{2}{3}n}$) decreasing TUF accuracy $\varepsilon$ can be reached with sample complexity*

$$\mathcal{O}\Big( \frac{1}{\xi^2(1-\gamma)^5\varepsilon^2} \Big).$$

*The $\mathcal{O}$ notation is asymptotic in $\varepsilon$, with no other constants in $|\mathcal{S}|$, $|\mathcal{A}|$, and $\gamma$.*

The theorem shows that there is a provable improvement in sample complexity if the cycle length $K$ increases like $\gamma^{-\frac{2}{3}n}$. Note that $\log(2(1+\gamma)^{-1})^{-1} \sim 2(1-\gamma)^{-1}$ as $\gamma \to 1$. Assuming a (theoretical) uniform state–action sampling distribution with $\xi = \frac{1}{|\mathcal{S}||\mathcal{A}|}$, the leading constant in the sample complexity exhibits a quadratic dependence on $|\mathcal{S}||\mathcal{A}|$.

While the Q-learning complexity results are worst case (they hold for any MDP) and the logarithmic $\varepsilon$-improvement might appear marginal, the reality looks more interesting. In the bottom row of Figure 1 smoothed learning curves are plotted for our decreasing TUFs. It turns out that training stabilizes substantially. Decreasing TUF helps for high initial TUF, but does not harm low initial TUF.

*Remark* 1.2. Overestimating $Q$-values is a well-known issue in Q-learning. The reason why standard Q-learning (TUF=1) fails to converge is a significant overestimation at the beginning (see Figure 4 below) due to a large constant learning rate. From this perspective, optimizing TUF can be interpreted as additional overestimation reduction assistance due to better Bellman operator approximation.

The article is organized as follows. We start with a discussion of related work in Section 2. In Section 3 we collect basic notation and explain the cyclic structure of Q-learning with periodically updated target, sometimes called periodic Q-learning. Periodic Q-learning runs outer value-iterations $Q_{n+1} = T^* Q_n$ with inner SGD approximations of $T^* Q_n$. In Section 4, we analyze the errors accumulated in the inner and outer loops. Both are combined in Section 5, where we derive the optimal TUF design. Section 6 summarizes our analysis and provides an outlook for further research. Additional technical details and supplementary experiments are deferred to the appendix.

## 2. Related Work

The convergence properties of Q-learning and related algorithms have been studied extensively, beginning with early results on convergence under asynchronous stochastic approximation (Tsitsiklis, 1994). To name but a few complexity results, the generic lower bound $\Omega(\frac{|\mathcal{A}||\mathcal{S}|}{\varepsilon^2})$ of (Gheshlaghi Azar et al., 2013), the seminal upper bound $\tilde{\mathcal{O}}(\frac{(|\mathcal{A}||\mathcal{S}|)^{\frac{2\ln(1/\varepsilon)}{1-\gamma}}}{\varepsilon^2})$ of (Even-Dar & Mansour, 2003), and the $\tilde{\mathcal{O}}(\frac{|\mathcal{A}||\mathcal{S}|}{\varepsilon^4})$ complexity of phased Q-learning (Kearns & Singh, 1999).

The closest complexity results to ours are those of (Lee & He, 2020a) and (Chen et al., 2023). Lee & He (2020a) analyze periodic Q-learning with a fixed target update interval in the tabular setting and obtain a sample complexity of

order

$$\mathcal{O}\Big(\frac{|\mathcal{S}||\mathcal{A}|}{\epsilon^2(1-\gamma)^4 \log(\gamma^{-1})} \frac{L}{c^3} \log\big(\frac{1}{(1-\gamma)\epsilon}\big)\Big),$$

where $c$ and $L$ denote the minimal and maximal state-action sampling probabilities. Under a uniform state-action sampling distribution setting $\xi = c = L = (|\mathcal{S}|\,|\mathcal{A}|)^{-1}$, this becomes worse than our complexity bound under optimal TUF design $\mathcal{O}\big(\frac{1}{\xi^2(1-\gamma)^5\epsilon^2}\big)$ by a factor of order $|\mathcal{S}||\mathcal{A}| \log(\epsilon^{-1})$, up to logarithmic terms in $1 - \gamma$.[1] This improvement is due to two ingredients: a refined state-action level analysis of the asynchronous SGD inner loop, which already improves the complexity under fixed TUFs, and the optimal TUFs, which removes the additional logarithmic overhead in the target accuracy. Chen et al. (2023) study a more general fitted Q-iteration setting with target networks, Markovian sampling, and linear function approximation. Specializing their bound to the tabular case yields a complexity of order

$$\mathcal{O}\Big((t_\alpha + 1)\frac{\log\big(\frac{1}{\epsilon\sqrt{\lambda}(1-\gamma)^2}\big)\log\big(\frac{1}{(1-\gamma)\epsilon}\big)}{\lambda^3(1-\gamma)^4 \log(\gamma^{-1})}\epsilon^{-2}\Big),$$

where $\lambda$ corresponds to the minimal stationary state-action probability in the tabular case and $t_\alpha$ is a mixing-time term. Hence, under uniform sampling distribution (setting $\lambda = \xi$) and suppressing mixing-time effects, our bound again improves the dependence on the state-action space size and removes logarithmic factors in $\epsilon^{-1}$. We view these results as complementary: Chen et al. (2023) treat the more general Markovian and linear-function-approximation setting, whereas our contribution is an explicit optimal TUF design in the tabular asynchronous setting.

The interpretation of target freezing as a nested optimization scheme is closely related to classical approximate dynamic programming (ADP), where Bellman operators are applied inexactly. Error propagation under approximate Bellman updates has been studied extensively in value and policy iteration (Bertsekas & Tsitsiklis, 1996; Munos, 2005; Antos et al., 2008). Since the target network idea is younger than most results on ADP, it might come as no surprise that we cannot fit our analysis in standard results. In particular, our inner loop analysis is tailor-made for SGD optimization.

Besides target freezing, a natural way to reduce variance in $Q$-learning is to let the learning rate decrease over time; see e.g. (Even-Dar & Mansour, 2003). To the best of our knowledge, existing convergence results for asynchronous $Q$-learning that rely solely on diminishing learning rates as the variance-reduction mechanism are either qualitative, e.g. convergence results based on the Robbins-Monro conditions (Watkins & Dayan, 1992; Jaakkola et al., 1993; Tsitsiklis, 1994; Lee & He, 2020b), or yield sample complexities with

---

[1] Note that $\log(\gamma^{-1}) \sim (1-\gamma)^{-1}$ as $\gamma \to 1$.

an additional logarithmic factor in $\varepsilon^{-1}$, see, for instance, (Qu & Wierman, 2020; Chen et al., 2024). As a side-note, Li et al. (2023) consider the Ruppert–Polyak average of $Q$-learning with diminishing learning rates and show an upper bound $\tilde{O}(\frac{|\mathcal{A}||\mathcal{S}|}{\varepsilon^2})$ in the synchronous setting with no additional $\log(\varepsilon^{-1})$-factor.

Target networks were introduced in DQN to stabilize learning with function approximation (Mnih et al., 2015), and have since been refined to continuous control and actor-critic (Lillicrap et al., 2016; Fujimoto et al., 2018). Fan et al. (2020) provide a statistical analysis for DQN under the assumption that the underlying Bellman regression problem is solved exactly at each iteration. The mitigation of the overestimation bias using, for example, double Q-learning goes back to (Van Hasselt, 2010) and in combination with target network freezing in the deep setting to (Van Hasselt et al., 2016). While we do not provide theory in the intersection of both topics, this might be interesting to investigate further.

Our cyclic view and the decreasing TUFs are related to multi-timescale stochastic approximation (Borkar, 1997), where different components evolve at different speeds. The framework of fast-slow systems has recently been applied to the ADAM optimizer in supervised learning (Dereich & Jentzen, 2024; Dereich et al., 2025). Classical analyses of two-timescale algorithms are usually conducted in the asymptotic regime with fully separated timescales. In contrast, we derive an explicit and near-optimal growth schedule based on finite-time error bounds.

## 3. Preliminaries: From Basics to Target Update Frequencies

**Basics:** For the theoretical results, we consider an MDP defined by a tuple $(\mathcal{S}, \mathcal{A}, P, r, \gamma)$, where $\mathcal{S}$ is a finite state space, $\mathcal{A}$ a finite action space, $P : \mathcal{S} \times \mathcal{A} \times \mathcal{S} \rightarrow [0, 1]$ a transition probability function, $r : \mathcal{S} \times \mathcal{A} \rightarrow \mathbb{R}$ a reward function, and $\gamma \in [0, 1)$ a discount factor. A policy $\pi : \mathcal{S} \rightarrow [0, 1]^{\mathcal{A}}$ maps states to probability distributions over all actions, and the goal is to find an optimal policy $\pi^*$ that maximizes the expected discounted reward. The action-value function $Q^{\pi} : \mathcal{S} \times \mathcal{A} \rightarrow \mathbb{R}$ for a policy $\pi$ is defined as $Q^{\pi}(s, a) = \mathbb{E}[\sum_{t=0}^{\infty} \gamma^t r(s_t, a_t) \mid s_0 = s, a_0 = a, a_t \sim \pi(s_t)]$. The optimal action-value function $Q^*(s, a) = \max_{\pi} Q^{\pi}(s, a)$ can be characterized as the unique fixed point of the Bellman optimality operator $T^* : \mathbb{R}^{|\mathcal{S}| \times |\mathcal{A}|} \rightarrow \mathbb{R}^{|\mathcal{S}| \times |\mathcal{A}|}$ defined by

$$(T^*Q)(s, a) = \mathbb{E}_{s', r}\big[r(s, a) + \gamma \max_{a'} Q(s', a')\big].$$

The operator $T^*$ is a $\gamma$-contraction in the supremum norm, ensuring that repeated application $Q_{n+1} = T^*Q_n$ converges to its unique fixed point $Q^*$. The importance of

$Q^*$ stems from the fact that the greedy policy corresponding to $Q^*$ is optimal. For background theory we refer to (Puterman, 2014).

**Q-Learning, DQN, and Target Freezing:** The problem becomes significantly more challenging in the model-free setting, where only access to an estimator of $T^*$ is available, and in non-tabular regimes in which $|\mathcal{S}|$ and/or $|\mathcal{A}|$ are large or infinite. Approximate dynamic programming addresses this challenge by approximating $T^*$ and using parameterized function approximators $Q_\theta : \mathcal{S} \times \mathcal{A} \rightarrow \mathbb{R}$, often involving a neural network, the so-called $Q$-network. In that situation, the parameters $\theta \in \mathbb{R}^d$ are the learnable weights and biases of the neural network. The full tabular parameterization $Q_\theta(s, a) = \theta_{s,a}$ with $d = |\mathcal{S}| \cdot |\mathcal{A}|$ identifies the tabular situation as a special case. Rather than exactly computing $T^*Q_\theta$, ADP methods approximate the Bellman operator by minimizing a loss function. The general update uses the mean-squared Bellman error (MSBE):

$$\mathcal{L}(\theta)(s, a) = \frac{1}{2}\mathbb{E}_{s', r}\big[\big(Q_\theta(s, a) - \tau\big)^2\big],$$

with regression target $\tau = r(s, a) + \gamma \max_{a'} Q_\theta(s', a')$. An optimizer is then employed to iteratively update the parameters in the direction that reduces the MSBE. As an example, standard Q-learning

$$Q(s, a) \leftarrow Q(s, a) + \alpha\big(r + \gamma \max_{a'} Q(s', a') - Q(s, a)\big),$$

is the special case of MSBE minimization using the tabular parameterization and SGD with learning rate $\alpha$ for only one minimization step. It is well known that learning via optimizing MSBE is deeply troubling, see for instance (Riedmiller, 2005). The $Q$-network tends to overfit to present state-actions and hardly generalizes. To stabilize learning, Deep Q-Network (DQN) (Mnih et al., 2015) introduces two key modifications: an experience replay, which stores transitions in a buffer $\mathcal{D}$, and a target network with fixed parameters $\theta^-$. The modified loss function is given by

$$\overline{\mathcal{L}}(\theta) = \frac{1}{2}\mathbb{E}_{(s,a,r,s')\sim\mathcal{D}}\big[\big(Q_\theta(s, a) - \tau\big)^2\big], \quad (1)$$

where $\tau = r(s, a) + \gamma \max_{a'} Q_{\theta^-}(s', a')$. The target parameters $\theta^-$ are updated with $\theta$ every $K$ steps. For tabular Q-learning the target freezing results in the use of two matrices and updates

$$Q(s, a) \leftarrow Q(s, a) + \alpha\big(r + \gamma \max_{a'} Q^-(s', a') - Q(s, a)\big),$$

with target matrix $Q^-$ overwritten by $Q$ every $K$ steps.

**The Cyclic View on Target Freezing:** For our analysis we see periodic target freezing as a cyclic approach, using an *outer loop* for value iteration (Bellman updates) and an *inner loop* to approximate value iterates $T^*Q_n$ via SGD

(Q-learning updates with frozen target), summarized in the pseudocode (Algorithm 1). We denote by $K_n$ the number of inner optimization steps used for the $n$th step in the outer loop, also called the cycle length, and compare using only one inner optimization step $K_n = 1$ (Q-learning), constant $K_n = K$-steps (DQN), and, as we propose in this article, a geometrically increasing number $K_n$ (at rate $\gamma^{-\frac{2}{3}n}$) of optimization steps. Since $K_n$ is closely related to the DQN target update frequency, we call $1/K_n$ the *target update frequency* (TUF). Recall the simple fact from statistics that, for $X \in L^2(\Omega, \mathcal{F}, \mathbb{P})$, the scalar $\mathbb{E}[X]$ is the unique minimizer of the quadratic risk

$$\mathbb{E}[X] = \arg\min_{\theta \in \mathbb{R}} \mathbb{E}\big[(X - \theta)^2\big]. \qquad (2)$$

Thus, the computation of an expectation can be approached via *any* (stochastic) optimization algorithm applied to the loss $f(\theta) := \mathbb{E}[(X - \theta)^2]$. At the start of the $n$th Bellman update approximation the algorithm takes an outer iterate $Q_n$ and fixes it for the regression problem solved in the inner loop. The state-action pairs $(s_{n,k}, a_{n,k})$ are sampled according to a (possibly history-dependent) distribution $\pi_{n,k}$. For any state-action pair $(s, a)$, define the random variable $X_n(s, a) := r + \gamma \max_{a' \in \mathcal{A}} Q_n(s', a')$, where $(r, s') \sim p(\cdot \mid s, a)$ so that $(T^* Q_n)(s, a) = \mathbb{E}[X_n(s, a)]$. By (2), we obtain the equivalent regression characterization

$$(T^* Q_n)(s, a) = \arg\min_{\theta \in \mathbb{R}} \frac{1}{2} \mathbb{E}\big[\big(X_n(s, a) - \theta\big)^2\big]. \qquad (3)$$

To solve (3), a stochastic optimizer is run for $K_n$ steps using samples generated by the simulator. This produces a sequence of iterates $(Q_n^{(k)})_{k=0}^{K_n}$ according to

$$Q_n^{(k+1)} = \Psi_n^{(k)}(Q_n^{(k)}), \qquad (4)$$

where $\Psi_n^{(k)} : \mathbb{R}^{|\mathcal{S}| \times |\mathcal{A}|} \to \mathbb{R}^{|\mathcal{S}| \times |\mathcal{A}|}$ denotes a (random) measurable update operator induced by the stochastic optimizer at inner iteration $k$. The randomness in $\Psi_n^{(k)}$ arises from the simulator samples used at iteration $k$. This level of generality also allows us to model asynchronous update schemes, which will be instantiated concretely in Section 4.2. We represent the effect of the entire inner loop by a random operator $\widetilde{T}_n : \mathbb{R}^{|\mathcal{S}| \times |\mathcal{A}|} \to \mathbb{R}^{|\mathcal{S}| \times |\mathcal{A}|}$ defined implicitly through

$$\widetilde{T}_n Q_n^{(0)} := Q_n^{(K_n)}. \qquad (5)$$

$\widetilde{T}_n Q_n^{(0)}$ acts as an approximation of $T^* Q_n$. The next outer iterate is then initialized $Q_{n+1}^{(0)} := Q_n^{(K_n)}$. For the subsequent analysis, let $(\mathcal{F}_{n,k})_{n \geq 0, \, 0 \leq k \leq K_n}$ be the filtration generated by all samples and internal randomness up to inner iteration $k$ of the $n$th outer loop.

---

**Algorithm 1** Periodic Q-Learning (Tabular), Variable TUF

1: **Input:** Cycle lengths $\{K_n\}_{n=1}^N$
2: Initialize $Q : \mathcal{S} \times \mathcal{A} \to \mathbb{R}$ arbitrarily, $Q^- = Q$
3: **for** Outer Bellman iteration with $n = 1$ **to** $N$ **do**
4:     **for** Inner iteration with $k = 1$ **to** $K_n$ **do**
5:         Observe current state $s$
6:         Choose e.g. $a \sim \varepsilon$-greedy$(Q(s, \cdot))$
7:         Execute action $a$, observe $r, s'$
8:         Compute $\delta = r + \gamma \max_{a' \in \mathcal{A}} Q^-(s', a')$
9:         Update $Q(s, a) \leftarrow (1 - \alpha)Q(s, a) + \alpha \delta$
10:    **end for**
11:    $Q^- \leftarrow Q$ (update delayed target)
12: **end for**

---

## 4. Abstract Convergence Analysis

### 4.1. Outer Loop Analysis

We start with a general convergence result for the outer loop. For this, it suffices to quantify how accurately the inner loop approximates the Bellman operator. We define the (conditional) operator approximation error

$$\eta_n := \mathbb{E}\Big[\big\|\widetilde{T}_n Q_n^{(0)} - T^* Q_n^{(0)}\big\|_\infty \,\Big|\, \mathcal{F}_{n,0}\Big], \qquad (6)$$

which will be the main object of our error analysis for the inner loop. Given these approximation errors $(\eta_n)$, we derive a recursive estimate for the outer loop error. Importantly, the analysis in this section is based solely on the sequence of operator approximation errors $(\eta_n)$ and makes no assumptions about how the MSBE is minimized in the inner loop. It is a general framework that can be applied to any MSBE optimization method (e.g., SGD, Adam, RMSProp).

**Proposition 4.1** (Approximate contraction of the outer loop)**.** *Let $T^*$ be the Bellman optimality operator on $\mathbb{R}^{|\mathcal{S}| \times |\mathcal{A}|}$ and let $Q^*$ denote its unique fixed point. Let $\{Q_n^{(0)}\}_{n \geq 0}$ be generated by (4)–(5), and define*

$$E_n := \|Q_n^{(0)} - Q^*\|_\infty. \qquad (7)$$

*Then, for all $n \geq 0$,*

$$\mathbb{E}[E_{n+1} \mid \mathcal{F}_{n,0}] \leq \gamma E_n + \eta_n, \qquad (8)$$

*where $\eta_n$ is defined in (6). Moreover, if $\sum_{n=0}^{\infty} \eta_n < \infty$ almost surely, then $E_n \to 0$ almost surely as $n \to \infty$.*

*Proof.* Using the fixed-point identity $T^* Q^* = Q^*$, the triangle inequality, and the contraction property, gives

$$\begin{aligned}
E_{n+1} &= \|Q_{n+1}^{(0)} - Q^*\|_\infty \\
&= \|\tilde{T}_n^{(K_n)} Q_n^{(0)} - T^* Q^*\|_\infty \\
&\leq \|\tilde{T}_n^{(K_n)} Q_n^{(0)} - T^* Q_n^{(0)}\|_\infty + \|T^* Q_n^{(0)} - T^* Q^*\|_\infty \\
&\leq \|\tilde{T}_n^{(K_n)} Q_n^{(0)} - T^* Q_n^{(0)}\|_\infty + \gamma \|Q_n^{(0)} - Q^*\|_\infty.
\end{aligned}$$

Taking conditional expectation given $\mathcal{F}_{n,0}$ yields (8). Note that $(E_n)$ and $(\eta_n)$ are both non-negative and $(\mathcal{F}_{n,0})$-adapted. For the almost sure convergence claim, apply an adaptation of the Robbins–Siegmund theorem, provided in Corollary 4.2 below, to the nonnegative adapted process $\{E_n\}_{n\geq 0}$ using (8) and the assumption $\sum_{n\geq 0}\eta_n < \infty$ almost surely. $\square$

The above proof relies on the important adaptation of the Robbins–Siegmund theorem (Robbins & Siegmund, 1971), a classical tool in the analysis of stochastic approximation methods; see, for example, Corollary B.7 in (Kassing et al., 2025).

**Corollary 4.2.** *Let $(\Omega, \mathcal{A}, \mathcal{F}, \mathbb{P})$ be a filtered probability space, $(Z_k)_{k\in\mathbb{N}}$, $(A_k)_{k\in\mathbb{N}}$, $(B_k)_{k\in\mathbb{N}}$ and $(D_k)_{k\in\mathbb{N}}$ be non-negative and $\mathcal{F}$-adapted stochastic processes such that*

$$\sum_{k=0}^{\infty}A_k < \infty, \quad \sum_{k=0}^{\infty}B_k < \infty \quad and \quad \sum_{k=0}^{\infty}D_k = \infty$$

*almost surely. Moreover, suppose*

$$\mathbb{E}[Z_{k+1} \mid \mathcal{F}_k] \leq Z_k(1 + A_k - D_k) + B_k.$$

*Then $Z_k$ converges almost surely to $0$ for $k \to \infty$.*

With Proposition 4.1 in hand we can clearly separate the effect of contraction and approximation accuracy:

(i) the *outer loop contraction* is governed solely by $\gamma$,

(ii) the *inner loop approximation error* $\eta_n$, which depends on the sampling process, cycle length $K_n$, and the chosen optimizer.

### 4.2. Inner Loop: Analysis of SGD Iterates

In this section, we specialize the abstract framework of Section 4.1 to the case where the inner loop optimizer for the MSBE is SGD. We derive non-asymptotic error bounds on the accuracy with which each inner loop approximates a Bellman update under asynchronous sampling.

**Assumption 4.3.**
- Persistent exploration: In every step each state-action pair is chosen with probability at least $\xi \in (0, 1]$. The state-action exploration distribution $\pi_{n,k}$ is allowed to be history-dependent.
- Reward variances are bounded by some $\sigma^2 > 0$.

*Remark* 4.4. To have an example in mind, the reader might think of $\varepsilon$-greedy or uniform exploration. One may replace the persistent exploration assumption by some weaker sampling conditions. For instance, one may assume that, on average, each state-action pair is visited at least once within a finite time window, an assumption that can be used in the analysis of asynchronous stochastic approximation schemes (Beck & Srikant, 2012). This condition is stronger

than merely requiring infinitely many updates to each state–action pair, as considered for example in (Yu & Bertsekas, 2013), but weaker than uniform state-action visitation.

Fix an index $n$ and cycle length $K_n$. As discussed, the Bellman target is fixed at the beginning of the inner loop and defined by $Q_n^{(0)}$. At inner iteration $k$, a state-action pair $(s_{n,k}, a_{n,k})$ is sampled from $\pi_{n,k}$, and the tabular Q-matrix is updated asynchronously as

$$Q_n^{(k+1)}(s,a) = \Psi_n^{(k)}(Q_n^{(k)}, s_{n,k}, a_{n,k})(s,a) := Q_n^{(k)}(s,a)$$
$$+ \mathbb{1}_{\{(s,a)=(s_{n,k},a_{n,k})\}}\alpha_n^{(k)}((\hat{T}Q_n^{(0)})(s,a) - Q_n^{(k)}(s,a)),$$

where for $(r_{n,k}, s'_{n,k}) \sim p(\cdot \mid s_{n,k}, a_{n,k})$

$$(\hat{T}Q_n^{(0)})(s,a) = r_{n,k} + \gamma\max_{a'\in\mathcal{A}}Q_n^{(0)}(s'_{n,k},a')$$

is an unbiased estimator of the Bellman optimality operator $(T^*Q_n^{(0)})(s,a)$ conditionally on $(s_{n,k}, a_{n,k}) = (s,a)$. After completing $K_n$ inner steps, the next outer iterate is set to $Q_{n+1}^{(0)} := Q_n^{(K_n)}$.

We define the iterating inner loop approximation error

$$e_n^{(k)} := \mathbb{E}[\|Q_n^{(k)} - T^*Q_n^{(0)}\|_2^2 \mid \mathcal{F}_{n,0}].$$

To obtain an explicit convergence rate, we now specialize to a decreasing learning rate schedule of order $\frac{1}{k}$ which is a common schedule for SGD on quadratic objectives and gives the fastest rate of convergence (Jentzen & von Wurstemberger, 2020).

**Theorem 4.5** (Inner loop convergence rate). *Suppose Assumption 4.3 holds and choose $\alpha_n^{(k)} = \frac{2}{\xi(k+\frac{2}{\xi})}$. Then, for all $k \geq 0$,*

$$e_n^{(k+1)} \leq \frac{c_1\|Q_n^{(0)} - Q^*\|_\infty^2 + c_2}{k + \frac{2}{\xi}}, \tag{9}$$

*where*

$$c_1 := \left(\frac{2}{\xi} + 1\right)|\mathcal{S}||\mathcal{A}|(1+\gamma)^2 + \left(\frac{16}{\xi^2} + \frac{8}{\xi}\right)\gamma^2,$$
$$c_2 := \left(\frac{8}{\xi^2} + \frac{4}{\xi}\right)(\sigma^2 + 2\gamma^2\|Q^*\|_\infty^2).$$

The complete proof is deferred to Appendix B.1. In Proposition B.1, we first employ standard stochastic approximation arguments to derive a one-step recursion for the inner loop error of the form

$$e_n^{(k+1)} \leq (1 - \xi\alpha_n^{(k)})e_n^{(k)}$$
$$+ (\alpha_n^{(k)})^2(2\sigma^2 + 4\gamma^2 e_n^{(0)} + 4\gamma^2\|Q^*\|_\infty^2), \tag{10}$$

valid for arbitrary learning rates $\alpha_n^{(k)} \in (0, 1]$. The main technical challenge is to control the asynchronous update

mechanism, under which only a single state–action pair is updated at each iteration. This leads to an effective contraction of order $(1 - \xi \alpha_n^{(k)})$, reflecting the minimum visitation probability $\xi$. In a second step, we show by induction that for the learning rate schedule $\alpha_n^{(k)} = \frac{2}{\xi(k + \frac{2}{\xi})}$ the recursion (10) yields the inner loop convergence bound stated in (9).

### 4.3. Almost Sure Convergence

Theorem 4.5 shows that the conditional Bellman operator approximation error behaves as

$$\eta_n \leq \frac{\sqrt{c_1} E_n + \sqrt{c_2}}{\sqrt{K_n}}.$$

In combination with Proposition 4.1 we deduce that

$$\mathbb{E}\big[E_{n+1} \mid \mathcal{F}_{n,0}\big] \leq \gamma E_n + \frac{\sqrt{c_1} E_n + \sqrt{c_2}}{\sqrt{K_n}}. \qquad (11)$$

If the inner loop errors are summable, the outer loop contraction dominates, and we get convergence. The proof of the next result is deferred to Appendix B.2.

**Corollary 4.6.** *Suppose that $K_n$ satisfies $\sum_{n=0}^{\infty} \frac{1}{\sqrt{K_n}} < \infty$. Then, under the setting of Theorem 4.5, we have $\|Q_n^{(0)} - Q^*\|_\infty \to 0$ almost surely as $n \to \infty$.*

## 5. Optimal TUF Design

In this section, we explore how to choose the TUF $(1/K_n)$ with the aim of minimizing the total computational cost required to achieve a certain target accuracy. This task can be interpreted as a multilevel optimization problem, inspired by multilevel Monte–Carlo methods (Giles, 2008) and more recently applied to inverse problems in Weissmann et al. (2022), whose proof techniques are closely related to our analysis. In the following, we assume that the computational cost of obtaining the iterate $Q_n^{(0)}$ is proportional to the number of sampled state–action pairs. Specifically, we measure the cost by

$$\mathrm{cost}(Q_n^{(0)}) = \mathrm{cost}(Q_{n-1}^{(K_{n-1})}) := \sum_{j=0}^{n-1} K_j$$

which we refer to as the sample complexity of $Q_n^{(0)}$. Given an accuracy of $\varepsilon > 0$, our overall goal is to select $N$ and $(K_n)_{n=0}^{N-1}$ solving the constrained optimization problem

$$\min_{N,(K_n)_{n=0}^{N-1}} \mathrm{cost}(Q_N^{(0)}) \quad \text{s.t.} \quad e_N := \mathbb{E}[E_N] \leq \varepsilon. \quad (12)$$

To this end, we introduce an effective contraction factor $\mu \in (\gamma, 1)$ and enforce the TUF to be sufficiently low (and

hence the cycle length $K_n$ sufficiently large) such that for all $n \in \mathbb{N}_0$,

$$0 < \gamma + \sqrt{\frac{c_1}{K_n}} \leq \mu < 1. \qquad (13)$$

More precisely, we require $\min_n K_n \geq K_{\min}(\mu) := \frac{c_1}{(\mu - \gamma)^2}$. Thus, setting $e_n := \mathbb{E}[E_n]$ for $n = 0, \dots, N$ and taking expectation on both sides in Equation (11) and iterating over $n \in \mathbb{N}$ yields

$$e_n \leq \mu^n e_0 + \sum_{j=0}^{n-1} \mu^{n-1-j} \sqrt{\frac{c_2}{K_j}}, \quad n \in \mathbb{N}. \qquad (14)$$

In the subsequent analysis, we will enforce $e_N \leq \varepsilon$ by decomposing via (14) and requiring that

$$\mu^N e_0 \leq \frac{\varepsilon}{2} \quad \text{and} \quad \sum_{j=0}^{N-1} \mu^{N-1-j} \sqrt{\frac{c_2}{K_j}} \leq \frac{\varepsilon}{2}$$

which fixes the number of outer loops to $N(\varepsilon) \geq \left\lceil \frac{\log(\varepsilon/(2e_0))}{\log(\mu)} \right\rceil$ and reduces the design problem to selecting $K_j(\varepsilon)$, $j = 0, \dots, N(\varepsilon) - 1$. Therefore, we consider so-called quasi-optimal (fixed/variable) TUF schedules (see Definition C.1) which cannot be improved asymptotically as $\varepsilon \to 0$ (e.g. by splitting the error terms in (14) asymmetrically). For simplicity, we will allow the cycle lengths $(K_j)$ to be real-valued.

**Fixed TUF.** We begin by considering the simplest design in which the number of inner SGD steps is held constant across the inner loops. In this case, we end up with minimizing $\mathrm{cost}(Q_N^{(0)}) = NK$ under the constraint $e_N \leq \varepsilon$. The following theorem quantifies the resulting trade-off between the number of outer iterations and the cost per inner loop. The full proof is given in Appendix C.

**Theorem 5.1** (Fixed TUF). *Set $\mu = \mu(\gamma) = \frac{1+\gamma}{2}$. Then the choice*

$$N(\varepsilon) = \left\lceil \frac{\log(\frac{\varepsilon}{2e_0})}{\log(\mu)} \right\rceil, \quad K(\varepsilon) = \frac{4c_2}{\varepsilon^2} \cdot \left(\frac{1 - \mu^{N(\varepsilon)}}{1 - \mu}\right)^2$$

*defines a quasi-optimal fixed TUF. If the rewards are bounded by $1$ it holds*

$$\mathrm{cost}(Q_{N(\varepsilon)}^{(0)}) = \mathcal{O}\left(\frac{\log((1-\gamma)^{-1}\varepsilon^{-1})}{(1-\gamma)^4 \xi^2 \log(2(1+\gamma)^{-1})\varepsilon^2}\right).$$

The sample complexity for the fixed TUF exhibits a logarithmic dependence on $\varepsilon$ arising from the contraction of the Bellman operator, and a quadratic dependence on $\varepsilon$ induced by the approximation of the Bellman operator in the inner loop.

*Remark* 5.2. Note that the leading constant of the sample complexity under fixed TUF scales by

$$\frac{1}{(1-\gamma)^4 \log(2(1+\gamma)^{-1})} \sim \frac{1}{(1-\gamma)^5}$$

asymptotically as $\gamma \to 1$. Moreover, the above result shows that the choice $N(\varepsilon), K(\varepsilon)$ defines a quasi-optimal fixed TUF. This is verified in the proof by showing that any pair $\hat{N}, \hat{K}$ satisfying

$$\mu^{\hat{N}} e_0 + \sqrt{\frac{c_2}{\hat{K}}} \sum_{j=0}^{\hat{N}} \mu^{N-1-j} \leq \varepsilon$$

must obey

$$N(2\varepsilon)K(2\varepsilon) \leq \hat{N}\hat{K} .$$

Consequently, under the error bound (14), the fixed TUF complexity cannot be improved in terms of its dependence on $1 - \gamma, \xi$ and $\varepsilon$.

**Optimal TUF Design.** The fixed TUF treats all outer iterations equally, despite the fact that their roles differ significantly. Early in training, when $e_n$ is large, the contraction term $\gamma e_n$ dominates the dynamics, and only a coarse approximation of the Bellman operator is required. Conversely, in later stages of learning, the iterates approach the fixed point, and progress is limited by the accuracy with which the Bellman operator is approximated. This observation suggests that computational effort should be distributed *non-uniformly* across inner loops: short inner loops are sufficient early on, while increasingly accurate (and hence longer) inner loops are required to reach high accuracy. We now formalize the intuition by optimizing the sequence of cycle lengths $(K_n)$ under a fixed target accuracy $\varepsilon$. Our final theorem provides an explicit construction of a decreasing TUF schedule that solves the cost minimization problem (12). We provide a detailed proof in Appendix C.

**Theorem 5.3.** *Set* $\mu = \mu(\gamma) = \frac{1+\gamma}{2}$. *Then the choice*

$$N(\varepsilon) = \left\lceil \frac{\log(\frac{\varepsilon}{2e_0})}{\log(\mu)} \right\rceil, \quad K_j(\varepsilon) = C_\varepsilon \mu^{\frac{2}{3}(N(\varepsilon)-1-j)},$$
$$C_\varepsilon = \frac{4c_2}{\varepsilon^2} \left( \frac{1 - \mu^{\frac{2}{3}N(\varepsilon)}}{1 - \mu^{\frac{2}{3}}} \right)^2, \tag{15}$$

*for* $j = 0, \ldots, N(\varepsilon) - 1$ *defines a quasi-optimal TUF schedule. If the rewards are bounded by 1 it holds*

$$\text{cost}(Q_{N(\varepsilon)}^{(0)}) = \mathcal{O}\left( \frac{1}{\xi^2(1-\gamma)^5\varepsilon^2} \right).$$

The theorem also shows that choosing the cycle length to increase geometrically at rate $\gamma^{-\frac{2}{3}n}$ improves the sample complexity by removing the logarithmic factor $|\log(\frac{1}{(1-\gamma)\varepsilon})|$.

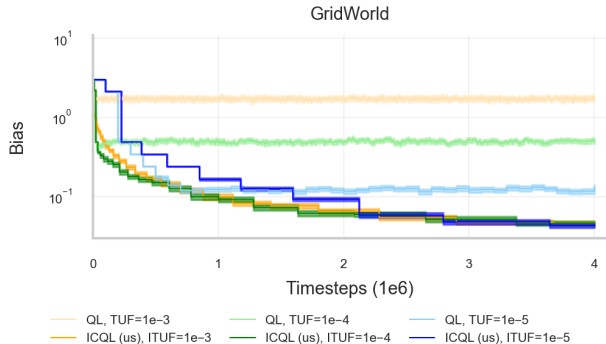

*Figure 2.* Q-learning on GridWorld from Appendix A.2 with optimally decreasing TUFs from Theorem 5.3 in comparison to Q-learning with fixed TUF for $K \in \{1000, 10000, 100000\}$. ITUF denotes the initial TUF in our ICQL algorithm. Shaded bands indicate uncertainty 95% confidence intervals across 50 random seeds: High fixed TUFs lead to early plateaus due to coarse Bellman operator approximations, while low fixed TUFs converge slowly because overly accurate inner-loop approximations dominate early iterations. Geometrically decreasing TUF schedule predicted by our theory avoids this trade-off by balancing contraction and Bellman approximation error, achieving both rapid initial progress and asymptotic convergence.

The geometrically increasing cycle lengths arise naturally from the structure of the error recursion. Recall that, after $N$ outer iterations, the error satisfies

$$e_N \leq \mu^N e_0 + \sum_{j=0}^{N-1} \mu^{N-1-j} \sqrt{\frac{c_2}{K_j}},$$

showing that the approximation error generated in cycle $j$ is discounted by the factor $\mu^{N-1-j}$. Consequently, inaccuracies produced in early cycles have substantially less influence on the final error than inaccuracies produced in later cycles. Using a fixed TUF, therefore, allocates computational effort inefficiently, solving early inner problems too accurately and introducing an additional logarithmic overhead. In contrast, the optimal variable schedule balances the discounted approximation errors across cycles by increasing $K_j$ geometrically.

Although this constitutes only a logarithmic improvement over fixed target update schedules from a worst-case theoretical perspective, the numerical experiments below demonstrate that the resulting gains can be substantial in practice. As an illustration of the theorem, Figure 2 compares different constant TUFs with the optimal decreasing TUF. The plot shows the bias $\|Q_n - Q^*\|_\infty$, which converges rapidly to zero under the optimal schedule. We observe that constant TUF can decrease the bias fast, but ultimately saturates. In contrast, decreasing TUF can decrease equally fast without saturation (caused by the increasingly precise Bellman approximation).

*Remark* 5.4. Similarly as in the fixed TUF case, to prove that the choice $N(\varepsilon), K_j(\varepsilon), j = 1, \ldots, N(\varepsilon) - 1$ is quasi-

optimal we derive that any choice $\hat{N}$, $\hat{K}_j$, $j = 0, \ldots, \hat{N} - 1$ satisfying

$$\mu^{\hat{N}} e_0 + \sum_{j=0}^{\hat{N}-1} \mu^{\hat{N}-1-j} \sqrt{\frac{c_2}{\hat{K}_j}} \leq \varepsilon$$

must obey

$$\sum_{j=0}^{N(2\varepsilon)-1} K_j(2\varepsilon) \leq \sum_{j=0}^{\hat{N}-1} \hat{K}_j \, .$$

Hence, under the error bound (14) the variable TUF complexity cannot be improved in terms of its dependence on $1 - \gamma$, $\xi$ and $\varepsilon$.

For convenience of the reader we added a detailed formulation of the resulting algorithm in Appendix A.1. We call the algorithm *Increasing-Cycle Q-Learning* (ICQL) as it uses the theoretically optimal geometrically increasing cycle lengths and hence decreasing TUFs. The decreasing TUF idea can be integrated readily into standard DQN implementations, as we did for the Lunar Lander simulations in Figure 1.

## 6. Contributions and Future Work

This article builds on the principled approximate dynamic programming view of target networks, interpreting periodic target fixing as a nested Bellman approximation scheme. We show that the TUF controls the bias-variance trade-off in Bellman operator approximation and governs convergence behavior. For tabular Q-learning with fixed targets, we derive refined non-asymptotic convergence bounds under asynchronous sampling and SGD updates of the inner loop (the mean squared Bellman error minimization). Our analysis reveals that using a fixed TUF is provably suboptimal. We derive an explicit geometrically increasing cycle lengths schedule of order $\gamma^{-\frac{2}{3}n}$ that is near-optimal. The resulting schedule removes the logarithmic $\varepsilon$-dependence in sample complexity compared to the best fixed-TUF choice. In particular, we obtain an $\mathcal{O}((1 - \gamma)^{-5}\xi^{-2}\varepsilon^{-2})$ sample complexity bound for tabular Q-learning with optimally decreasing TUFs. As an illustration of the results we consider a GridWorld and Lunar Lander with DQN using SGD error minimization (to fit the theory). It turns out that, in line with heuristic findings of previous articles, decreasing TUF helps stabilize the learning process.

This theory article was motivated by a concrete practical problem: the choice of target network update frequencies in Q-learning and DQN, and the lack of theoretical guidance for this design decision. Our work opens several directions for future research. (i) To further bridge the gap to modern deep reinforcement learning practice, it is necessary to move beyond SGD and analyze inner-loop optimization with adaptive methods such as ADAM. In this work, the inner loop was modeled as SGD applied to a quadratic objective arising from Bellman regression. Extending the analysis to ADAM therefore requires a theoretical understanding of its behavior on (stochastic) quadratic objectives, which remains an active area of research. (ii) In the absence of a theoretical analysis, it is of practical interest to investigate whether the proposed approach remains effective when combined with adaptive optimization methods such as ADAM as well. Furthermore, our approach may be applicable beyond Q-learning and DQN to other ADP algorithms that employ target network freezing. Allocating fewer inner-loop iterations to earlier updates where the contraction dominates the approximation errors should still be advantageous. Preliminary experimental results in these directions are provided in Appendix D.1. (iii) The approximate Bellman contraction perspective considered in this paper naturally suggests extensions beyond predetermined TUF schedules. The outer-loop recursion in Proposition 4.1 shows that progress depends not directly on the number of inner iterations, but on the resulting Bellman approximation accuracy $\eta_n$. This observation motivates *accuracy-triggered target updates*, in which the target network is updated as soon as the inner loop achieves a prescribed approximation accuracy, rather than after a fixed number of steps. Concretely, instead of fixing the cycle lengths $K_n$ in advance, one specifies a sequence of accuracy thresholds for $\eta_n$ and terminates the inner loop once the desired accuracy is reached. In the tabular setting, we outline a practical mechanism for estimating $\eta_n$ online during the inner loop, enabling such early stopping rules without additional oracle access. Details of this accuracy-triggered scheme, along with a first illustrative GridWorld experiment, are provided in Appendix D.2.

## Code

The code used in our experiments can be found on GitHub: https://github.com/BommeHD/ICQL.git. For long-term archival stability the codebase has been permanently deposited on Zenodo under DOI 10.5281/zenodo.20287291 (Weissmann et al., 2026).

## Acknowledgments

The authors acknowledge support by the state of Baden-Württemberg through bwHPC and the German Research Foundation (DFG) through grant INST 35/1597-1 FUGG as well as project CRC/TRR 388 "Rough Analysis, Stochastic Dynamics and Related Fields" – Project ID 516748464. Furthermore, the authors thank Tyler Clark for sharing his code for the algorithm BTR+RISE.

## Impact Statement

This paper presents work whose goal is to advance the field of machine learning. There are many potential societal consequences of our work, none of which we feel must be specifically highlighted here.

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

# A. Algorithmic Details

## A.1. Increasing-Cycle Q-Learning

In this section, we present the concrete algorithmic instantiation using SGD in the inner loop. This setting corresponds to classical periodic Q-learning in the tabular case. Motivated by the refined convergence analysis of Sections 4 and 5, we propose to increase the number of SGD updates performed between target synchronizations over time. Early inner loops use short target-freezing periods to exploit the contraction of the Bellman operator efficiently, while later inner loops gradually decrease the TUF to reduce approximation error and avoid convergence saturation.

The resulting algorithm, termed *Increasing-Cycle Q-Learning* (ICQL), differs from standard periodic Q-learning only in the choice of the cycle lengths $(K_n)$, which are grown geometrically according to the theoretically optimal schedule derived in Theorem 5.3.

---

**Algorithm 2** Increasing-Cycle Q-Learning (ICQL)

---

1: **Input:** Initial $Q$-matrix $Q_0^{(0)}$; discount factor $\gamma \in (0,1)$; initial cycle length $K_0$; number of target updates $N$; learning rate schedule $\alpha_n^{(k)}$
2: **Output:** $Q_T^{(0)} \approx Q^*$
3: **for** $n = 0, 1, \ldots, N-1$ **do**
4:      Set cycle length $K_n = \left\lceil K_0 \gamma^{-\frac{2}{3}n} \right\rceil$
5:      Initialize inner iterate $Q_n^{(0)}$
6:      **for** $k = 0, 1, \ldots, K_n - 1$ **do**
7:          Observe current state $s$
8:          Choose e.g. $a \sim \varepsilon\text{-greedy}(Q_n^{(k)}(s, \cdot))$
9:          Execute action $a$, observe $r, s'$
10:         Compute Bellman target: $\tau \leftarrow r + \gamma \max_{a'} Q_n^{(0)}(s', a')$
11:         Update $Q_n^{(k+1)}(s, a) \leftarrow Q_n^{(k)}(s, a) + \alpha_n^{(k)}(\tau - Q_n^{(k)}(s, a))$
12:      **end for**
13:      Set $Q_{n+1}^{(0)} \leftarrow Q_n^{(K_n)}$
14: **end for**

---

## A.2. GridWorld

| State Type | Reward Distribution | Mean | Variance |
|---|---|---|---|
| Start & Default | $\{-0.08, 0.05\}, p = 0.5$ | $-0.015$ | $0.0018225$ |
| Goal | $\{0.5, 1.5\}, p = 0.5$ | $1.0$ | $0.25$ |
| Stochastic Region | $\{-2.1, 2.0\}, p = 0.5$ | $-0.05$ | $4.2025$ |
| Bomb | deterministic: $-3$ | $-3$ | $0$ |

*Figure 3.* Grid World environment used for numerical experiments. The agent starts at state S (gray) and aims to reach the goal state G (green) while avoiding the bomb states B (red). The orange states represent the high variance region. The optimal path is indicated by the gray arrows.

We evaluate the proposed ICQL in a stochastic $4 \times 4$ GridWorld (Figure 3). The GridWorld environment consists of 52 state-action pairs, since taking an action, which would lead to the agent leaving the grid, leads to hovering in place. Rewards are received upon leaving each state. Each step in the grid incurs a small negative expected reward, encouraging the agent to find the shortest path to the goal. The difficulties lie in the high variance of the rewards in this region, which can mislead the learning algorithm into overestimating the value of actions leading into it. The optimal paths avoid the high variance region

because the expected reward there is lower than that of the default states, despite the potential for high positive rewards. During evaluation, the agent starts in the top left cell and does 7 steps, which is just enough to reach and further leave the goal state to collect the rewards. The optimal Q-value at the start state with action $a \in \{\text{down}, \text{right}\}$ is 0.0735 for discount factor set to $\gamma = 0.7$, 0.4615 for $\gamma = 0.9$ and 0.6551 for $\gamma = 0.95$. All experiments are conducted right in the setting of our

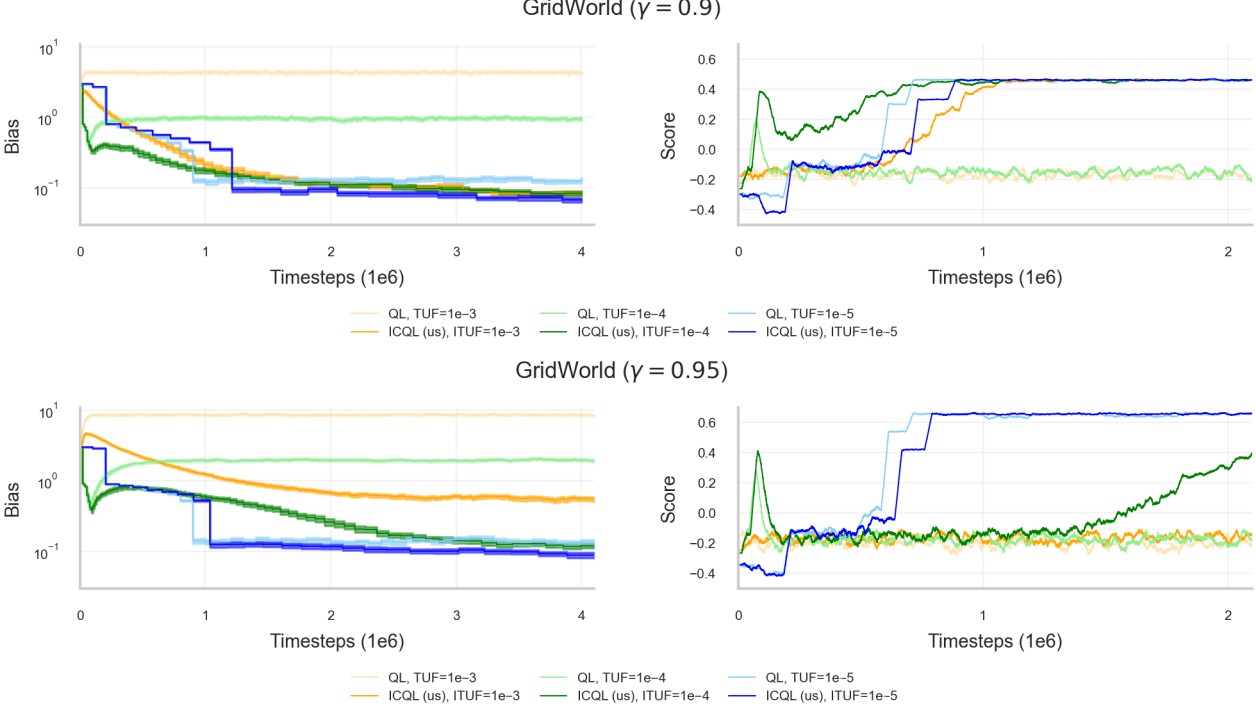

*Figure 4.* Q-learning on GridWorld from Appendix A.2 with optimally decreasing TUFs from Theorem 5.3 in comparison to Q-learning with fixed TUFs for $K \in \{1000, 10000, 100000\}$. The discount factor in the top row is set to $\gamma = 0.9$, below $\gamma = 0.95$. Shaded bands in the first column indicate uncertainty $95\%$ confidence intervals across 50 random seeds: High fixed TUFs lead to early plateaus due to coarse Bellman operator approximations, while low fixed TUFs converge slowly because overly accurate inner-loop approximations dominate early iterations. Geometrically decreasing TUF schedule predicted by our theory avoids this trade-off by balancing contraction and Bellman approximation error, achieving both rapid initial progress and asymptotic convergence, leading to the choice of the optimal path during evaluation.

theory. Exploration is done uniformly over all state-action pairs, leading to $\xi = 1/|\mathcal{S}||\mathcal{A}| = 1/52$. The learning rate for the multilevel version is set according to Theorem 4.5 by

$$\alpha_k = \frac{1}{1 + \frac{1}{2 \cdot 52}k}, \quad k = 0, 1, 2, \dots.$$

Figure 4 displays GridWorld results for $\gamma \in \{0.9, 0.95\}$ (top and bottom rows, respectively), showing both bias (left column) and the achieved score during evaluation (right column) for fixed-TUF Q-learning (QL) and the proposed increasing-cycle method (ICQL) with different initial target update frequencies (ITUF). Recall that the bias is defined as the $\ell_\infty$-distance between the cycle-start target matrix at each outer loop index $n$ to the optimal action-value function, i.e. $\text{Bias}(n) := \left\|Q_n^{(0)} - Q^*\right\|_\infty$. The GridWorld example from Figure 1 and Figure 2 relies on the discount factor set to $\gamma = 0.7$. For $\gamma = 0.9$, fixed-TUF QL exhibits pronounced schedule sensitivity: higher TUFs (e.g., $\text{TUF} = 10^{-3}, 10^{-4}$) lead to an early decrease followed by stabilization at a comparatively large error level, whereas only sufficiently low TUFs (here $\text{TUF} = 10^{-5}$) reach the small-bias regime around $10^{-1}$. ICQL decreases the bias more consistently and attains the lowest final errors across the compared settings. In particular, the staircase-like drops visible for lower ITUF reflect the cycle structure and indicate sustained improvement beyond the plateaus observed for fixed schedules. The score curves mirror these trends, since configurations that reach small bias attain high and stable returns, while the large-bias fixed-TUF runs remain trapped in a low-performance band.

The effect becomes more pronounced for $\gamma = 0.95$, where the weaker Bellman contraction amplifies target approximation

errors. Accordingly, fixed-TUF QL fails for TUF $= 10^{-3}$ (large bias throughout) and stabilizes at a large error for TUF $= 10^{-4}$, while TUF $= 10^{-5}$ is required to achieve small bias and high score. ICQL remains markedly more robust. While a high initialization (ITUF $= 10^{-3}$) can still exhibit substantial residual error, moderate to low initializations (ITUF $\in \{10^{-4}, 10^{-5}\}$) lead to continued bias reduction and corresponding improvements in score, with ITUF $= 10^{-5}$ matching the best-performing regime. Comparing to the $\gamma = 0.7$ bias plot in Figure 2, where all ICQL variants converge to low error while fixed-TUF QL typically plateaus at configuration-dependent levels, these results underpin that increasing the effective target-freezing horizon over training mitigates the stability–accuracy trade-off and yields improved robustness as the discount factor increases.

## B. Refined SGD Analysis: Proofs of Section 4

For the subsequent analysis we denote the history up to the $k$-th inner step of cycle $n$ as

$$\{Q\}_n^{(k)} := \{Q_i^{(j)} : i < n, \ 0 \le j \le K_i\} \cup \{Q_n^{(j)} : 0 \le j \le k\},$$
$$\{(s,a)\}_n^{(k)} := \{(s_{i,j}, a_{i,j}) : i < n, \ 0 \le j \le K_i\} \cup \{(s_{n,j}, a_{n,j}) : 0 \le j \le k\}.$$

and define the filtrations

$$\mathcal{F}_{n,k} := \sigma\big(\{Q\}_n^{(k)}, \ \{(s,a)\}_n^{(k-1)}\big),$$
$$\mathcal{F}_{n,k}^{s,a} := \sigma\big(\mathcal{F}_{n,k}, \ (s_{n,k}, a_{n,k}) = (s,a)\big).$$

Note that $\mathcal{F}_{n,0}$ contains all information up to the beginning of the $n$-th cycle. In particular, $Q_n^{(0)}$ is $\mathcal{F}_{n,0}$-measurable. Furthermore, $\mathcal{F}_{n,k}$ contains all information up to the $k$-th SGD step in the $n$-th cycle, but not the current sampled state–action pair $(s_{n,k}, a_{n,k})$. Conditioning on $\mathcal{F}_{n,k}^{s,a}$ fixes the current state–action pair $(s_{n,k}, a_{n,k}) = (s,a)$ and leaves randomness only in $(r_{n,k}, s'_{n,k})$. Consequently,

$$\mathbb{E}\Big[\hat{T} Q_n^{(0)}(s,a) \ \Big| \ \mathcal{F}_{n,k}^{s,a}\Big] = (T^* Q_n^{(0)})(s,a). \tag{16}$$

Define $I_{n,k}^{s,a} := \mathbb{1}_{\{(s_{n,k}, a_{n,k}) = (s,a)\}}$, which indicates whether the sampled state–action pair at step $k$ in inner loop $n$ equals $(s,a)$. It depends only on whether the current draw equals $(s,a)$, hence it is measurable with respect to $\mathcal{F}_{n,k}^{s,a}$. We allow the sampling law to use information contained in $\mathcal{F}_{n,k}$, such that the probability of sampling $(s,a)$ at step $k$ is given by $\pi_{n,k}(s,a)$, conditioned on the filtration $\mathcal{F}_{n,k}$. Since $(s_{n,k}, a_{n,k})$ is sampled according to $\pi_{n,k}$, the indicator function $\mathbb{1}_{\{(s_{n,k}, a_{n,k}) = (s,a)\}}$ takes the value 1 with probability $\pi_{n,k}(s,a)$ and 0 otherwise. Therefore,

$$\mathbb{E}\left[I_{n,k}^{s,a} \mid \mathcal{F}_{n,k}\right] = \pi_{n,k}(s,a). \tag{17}$$

### B.1. Inner Loop Convergence

We now analyze how well the inner SGD loop approximates a single Bellman update. The following proposition establishes a *one-step recursion* for the inner loop that cleanly separates a contraction term from a variance term induced by stochastic sampling.

**Proposition B.1** (One step inner loop recursion). *Suppose Assumption 4.3 holds, let $0 < \alpha_n^{(k)} \le 1$ and define*

$$e_n^{(k)} := \mathbb{E}\big[\|Q_n^{(k)} - T^* Q_n^{(0)}\|_2^2 \mid \mathcal{F}_{n,0}\big].$$

*Then for all $n \ge 0$, $k = 0, \dots, K_n - 1$ the inner loop error satisfies*

$$e_n^{(k+1)} \ \le \ \big(1 - \xi \alpha_n^{(k)}\big) e_n^{(k)} + (\alpha_n^{(k)})^2 C_n\,,$$

*where $C_n := 2\sigma^2 + 4\gamma^2\big(\|Q_n^{(0)} - Q^*\|_\infty^2 + \|Q^*\|_\infty^2\big)$.*

*Proof.* We define the per-coordinate error

$$\Delta_k(s,a) \ := \ Q_n^{(k)}(s,a) - (T^* Q_n^{(0)})(s,a)$$

and write

$$\mathbb{E}\left[\|Q_n^{(k+1)} - T^*Q_n^{(0)}\|_2^2 \mid \mathcal{F}_{n,0}\right] = \sum_{s,a} \mathbb{E}\left[(Q_n^{(k+1)}(s,a) - T^*Q_n^{(0)}(s,a))^2 \mid \mathcal{F}_{n,0}\right]$$
$$= \sum_{s,a} \mathbb{E}\left[\Delta_{k+1}(s,a)^2 \mid \mathcal{F}_{n,0}\right].$$

Using the inner loop update we have

$$Q_n^{(k+1)}(s,a) = Q_n^{(k)}(s,a) + \mathbb{1}_{\{(s,a)=(s_{n,k},a_{n,k})\}}\alpha_n^{(k)}((\hat{T}Q_n^{(0)})(s,a) - Q_n^{(k)}(s,a)).$$

Defining $\mathbb{1}_{\{(s,a)=(s_{n,k},a_{n,k})\}} = I_{n,k}^{s,a}$ the error term can be rewritten as

$$\Delta_{k+1}(s,a) = \Delta_k(s,a) + I_{n,k}^{s,a}\,\alpha_n^{(k)}\left((\hat{T}Q_n^{(0)})(s,a) - Q_n^{(k)}(s,a)\right)$$
$$= \Delta_k(s,a) + I_{n,k}^{s,a}\,\alpha_n^{(k)}\left((\hat{T}Q_n^{(0)})(s,a) - (T^*Q_n^{(0)})(s,a) - \Delta_k(s,a)\right).$$

Expanding the square, taking the conditional expectation with respect to $\mathcal{F}_{n,k}$ and applying $\mathcal{F}_{n,k}$-measurability of $\Delta_k(s,a)$ and $\alpha_n^{(k)}$, we obtain

$$\mathbb{E}\left[\Delta_{k+1}(s,a)^2 \mid \mathcal{F}_{n,k}\right] = \mathbb{E}\left[\Delta_k(s,a)^2 \mid \mathcal{F}_{n,k}\right]$$
$$+ 2\mathbb{E}\left[I_{n,k}^{s,a}\alpha_n^{(k)}\,\Delta_k(s,a)\left((\hat{T}Q_n^{(0)})(s,a) - (T^*Q_n^{(0)})(s,a) - \Delta_k(s,a)\right) \mid \mathcal{F}_{n,k}\right]$$
$$+ \mathbb{E}\left[I_{n,k}^{s,a}(\alpha_n^{(k)})^2\left((\hat{T}Q_n^{(0)})(s,a) - (T^*Q_n^{(0)})(s,a) - \Delta_k(s,a)\right)^2 \mid \mathcal{F}_{n,k}\right]$$
$$= \Delta_k(s,a)^2$$
$$+ 2\alpha_n^{(k)}\,\Delta_k(s,a)\mathbb{E}\left[I_{n,k}^{s,a}\left((\hat{T}Q_n^{(0)})(s,a) - (T^*Q_n^{(0)})(s,a) - \Delta_k(s,a)\right) \mid \mathcal{F}_{n,k}\right]$$
$$+ (\alpha_n^{(k)})^2\mathbb{E}\left[I_{n,k}^{s,a}\left((\hat{T}Q_n^{(0)})(s,a) - (T^*Q_n^{(0)})(s,a) - \Delta_k(s,a)\right)^2 \mid \mathcal{F}_{n,k}\right].$$

Now we treat each of the two conditional expectation terms separately and combine them afterwards:

Term 1: First, we further decompose the expectation by

$$\mathbb{E}\left[I_{n,k}^{s,a}\left((\hat{T}Q_n^{(0)})(s,a) - (T^*Q_n^{(0)})(s,a) - \Delta_k(s,a)\right) \mid \mathcal{F}_{n,k}\right]$$
$$= \underbrace{\mathbb{E}\left[I_{n,k}^{s,a}((\hat{T}Q_n^{(0)})(s,a) - (T^*Q_n^{(0)})(s,a)) \mid \mathcal{F}_{n,k}\right]}_{(a)} - \underbrace{\mathbb{E}\left[I_{n,k}^{s,a}\Delta_k(s,a) \mid \mathcal{F}_{n,k}\right]}_{(b)}.$$

For (a), we use the tower property with $\mathcal{F}_{n,k} \subset \mathcal{F}_{n,k}^{s,a}$ and use $\mathcal{F}_{n,k}^{s,a}$-measurability of $I_{n,k}^{s,a}$ to get

$$\mathbb{E}\left[I_{n,k}^{s,a}((\hat{T}Q_n^{(0)})(s,a) - (T^*Q_n^{(0)})(s,a)) \mid \mathcal{F}_{n,k}\right] = \mathbb{E}\left[\mathbb{E}\left[I_{n,k}^{s,a}((\hat{T}Q_n^{(0)})(s,a) - (T^*Q_n^{(0)})(s,a)) \mid \mathcal{F}_{n,k}^{s,a}\right] \mid \mathcal{F}_{n,k}\right]$$
$$= \mathbb{E}\left[I_{n,k}^{s,a}\mathbb{E}\left[((\hat{T}Q_n^{(0)})(s,a) - (T^*Q_n^{(0)})(s,a)) \mid \mathcal{F}_{n,k}^{s,a}\right] \mid \mathcal{F}_{n,k}\right].$$

By the unbiasedness of the one-sample Bellman target, we have

$$\mathbb{E}\left[((\hat{T}Q_n^{(0)})(s,a) - (T^*Q_n^{(0)})(s,a)) \mid \mathcal{F}_{n,k}^{s,a}\right] = 0,$$

such that (a) equals zero. For (b), we use the $\mathcal{F}_{n,k}$-measurability of $\Delta_k(s,a)$ and $\mathbb{E}\left[I_{n,k}^{s,a} \mid \mathcal{F}_{n,k}\right] = \pi_{n,k}(s,a)$ to derive

$$\mathbb{E}\left[I_{n,k}^{s,a}\Delta_k(s,a) \mid \mathcal{F}_{n,k}\right] = \Delta_k(s,a)\mathbb{E}\left[I_{n,k}^{s,a} \mid \mathcal{F}_{n,k}\right] = \Delta_k(s,a)\pi_{n,k}(s,a).$$

Combining (a) and (b), we have verified

$$\mathbb{E}\left[I_{n,k}^{s,a}\left((\hat{T}Q_n^{(0)})(s,a) - (T^*Q_n^{(0)})(s,a) - \Delta_k(s,a)\right) \mid \mathcal{F}_{n,k}\right] = -\Delta_k(s,a)\pi_{n,k}(s,a).$$

Term 2: We again split the expectation and apply the same arguments as before

$$\mathbb{E}\left[I_{n,k}^{s,a}\left((\hat{T}Q_n^{(0)})(s,a) - (T^*Q_n^{(0)})(s,a) - \Delta_k(s,a)\right)^2 \mid \mathcal{F}_{n,k}\right]$$

$$= \mathbb{E}\left[I_{n,k}^{s,a}((\hat{T}Q_n^{(0)})(s,a) - (T^*Q_n^{(0)})(s,a))^2 \mid \mathcal{F}_{n,k}\right]$$

$$- 2\mathbb{E}\left[I_{n,k}^{s,a}\Delta_k(s,a)((\hat{T}Q_n^{(0)})(s,a) - (T^*Q_n^{(0)})(s,a)) \mid \mathcal{F}_{n,k}\right]$$

$$+ \mathbb{E}\left[I_{n,k}^{s,a}\Delta_k(s,a)^2 \mid \mathcal{F}_{n,k}\right]$$

$$= \underbrace{\mathbb{E}\left[I_{n,k}^{s,a}((\hat{T}Q_n^{(0)})(s,a) - (T^*Q_n^{(0)})(s,a))^2 \mid \mathcal{F}_{n,k}\right]}_{(c)} + \underbrace{\mathbb{E}\left[I_{n,k}^{s,a}\Delta_k(s,a)^2 \mid \mathcal{F}_{n,k}\right]}_{(d)}.$$

For (c), we again use the tower property with $\mathcal{F}_{n,k} \subseteq \mathcal{F}_{n,k}^{s,a}$ and $\mathcal{F}_{n,k}^{s,a}$-measurability of $I_{n,k}^{s,a}$ to derive

$$\mathbb{E}\left[I_{n,k}^{s,a}((\hat{T}Q_n^{(0)})(s,a) - (T^*Q_n^{(0)})(s,a))^2 \mid \mathcal{F}_{n,k}\right] = \mathbb{E}\left[\mathbb{E}\left[I_{n,k}^{s,a}((\hat{T}Q_n^{(0)})(s,a) - (T^*Q_n^{(0)})(s,a))^2 \mid \mathcal{F}_{n,k}^{s,a}\right] \mid \mathcal{F}_{n,k}\right]$$

$$= \mathbb{E}\left[I_{n,k}^{s,a}\mathbb{E}\left[((\hat{T}Q_n^{(0)})(s,a) - (T^*Q_n^{(0)})(s,a))^2 \mid \mathcal{F}_{n,k}^{s,a}\right] \mid \mathcal{F}_{n,k}\right].$$

By expanding the square, we have

$$\mathbb{E}\left[((\hat{T}Q_n^{(0)})(s,a) - (T^*Q_n^{(0)})(s,a))^2 \mid \mathcal{F}_{n,k}^{s,a}\right] = \mathrm{Var}\left((\hat{T}Q_n^{(0)})(s,a) \mid \mathcal{F}_{n,k}^{s,a}\right)$$

$$= \mathrm{Var}\left(R(s,a) + \gamma \max_{a'} Q_n^{(0)}(S',a') \mid \mathcal{F}_{n,k}^{s,a}\right)$$

$$\leq 2\,\mathrm{Var}(r(s,a) \mid \mathcal{F}_{n,k}^{s,a}) + 2\gamma^2 \,\mathrm{Var}\left(\max_{a'} Q_n^{(0)}(S',a') \mid \mathcal{F}_{n,k}^{s,a}\right)$$

$$\leq 2\sigma^2 + 2\gamma^2 \|Q_n^{(0)}\|_\infty^2,$$

where we used $\mathrm{Var}(X + Y \mid \mathcal{F}_{n,k}^{s,a}) \leq 2\mathrm{Var}(X \mid \mathcal{F}_{n,k}^{s,a}) + 2\mathrm{Var}(Y \mid \mathcal{F}_{n,k}^{s,a})$ and the bound $\mathrm{Var}(Z \mid \mathcal{F}_{n,k}^{s,a}) \leq \mathbb{E}[Z^2 \mid \mathcal{F}_{n,k}^{s,a}] \leq \|Q_n^{(0)}\|_\infty^2$ for $Z := \max_{a'} Q_n^{(0)}(S',a')$. Hence (c) can be bounded by

$$\mathbb{E}\left[I_{n,k}^{s,a}((\hat{T}Q_n^{(0)})(s,a) - (T^*Q_n^{(0)})(s,a))^2 \mid \mathcal{F}_{n,k}\right] \leq \left(2\sigma^2 + 2\gamma^2\|Q_n^{(0)}\|_\infty^2\right)\pi_{n,k}(s,a).$$

For (d), we use the $\mathcal{F}_{n,k}$-measurability of $\Delta_k(s,a)^2$ and Equation (17) to deduce

$$\mathbb{E}\left[I_{n,k}^{s,a}\Delta_k(s,a)^2 \mid \mathcal{F}_{n,k}\right] = \Delta_k(s,a)^2\mathbb{E}\left[I_{n,k}^{s,a} \mid \mathcal{F}_{n,k}\right] = \Delta_k(s,a)^2\pi_{n,k}(s,a).$$

Combining (c) and (d), we end up with

$$\mathbb{E}\left[I_{n,k}^{s,a}\left((\hat{T}Q_n^{(0)})(s,a) - (T^*Q_n^{(0)})(s,a) - \Delta_k(s,a)\right)^2 \mid \mathcal{F}_{n,k}\right]$$

$$\leq \left(2\sigma^2 + 2\gamma^2\|Q_n^{(0)}\|_\infty^2\right)\pi_{n,k}(s,a) + \Delta_k(s,a)^2\pi_{n,k}(s,a).$$

Finally, combining Term 2 and Term 3, we obtain a recursion for the per-coordinate error:

$$\mathbb{E}\left[\Delta_{k+1}(s,a)^2 \mid \mathcal{F}_{n,k}\right] \leq \Delta_k(s,a)^2 - 2\alpha_n^{(k)}\,\pi_{n,k}(s,a)\,\Delta_k(s,a)^2$$

$$+ (\alpha_n^{(k)})^2\,\pi_{n,k}(s,a)\,\Delta_k(s,a)^2 + (\alpha_n^{(k)})^2\,\pi_{n,k}(s,a)\left(2\sigma^2 + 2\gamma^2\|Q_n^{(0)}\|_\infty^2\right)$$

$$= \left(1 - (2\alpha_n^{(k)} - (\alpha_n^{(k)})^2)\pi_{n,k}(s,a)\right)\Delta_k(s,a)^2 + (\alpha_n^{(k)})^2\,\pi_{n,k}(s,a)\left(2\sigma^2 + 2\gamma^2\|Q_n^{(0)}\|_\infty^2\right).$$

Define

$$E_n^{(k)} := \sum_{s,a} \Delta_k(s,a)^2 = \|Q_n^{(k)} - T^* Q_n^{(0)}\|_2^2.$$

Summing the per-coordinate bounds over all $(s,a)$, and using $\sum_{s,a} \pi_{n,k}(s,a) = 1$, $\pi_{n,k}(s,a) \geq \xi$, and $\alpha_n^{(k)} \leq 1$, we obtain

$$\begin{aligned}
\mathbb{E}[E_n^{(k+1)} \mid \mathcal{F}_{n,k}] &= \sum_{s,a} \mathbb{E}[\Delta_{k+1}(s,a)^2 \mid \mathcal{F}_{n,k}] \\
&\leq \sum_{s,a} \left(1 - (2\alpha_n^{(k)} - (\alpha_n^{(k)})^2)\pi_{n,k}(s,a)\right) \Delta_k(s,a)^2 + (\alpha_n^{(k)})^2 \sum_{s,a} \pi_{n,k}(s,a)\left(2\sigma^2 + 2\gamma^2 \|Q_n^{(0)}\|_\infty^2\right) \\
&= \sum_{s,a} \Delta_k(s,a)^2 - (2\alpha_n^{(k)} - (\alpha_n^{(k)})^2) \sum_{s,a} \pi_{n,k}(s,a)\Delta_k(s,a)^2 + (\alpha_n^{(k)})^2 \left(2\sigma^2 + 2\gamma^2 \|Q_n^{(0)}\|_\infty^2\right) \\
&\leq \left(1 - \xi(2\alpha_n^{(k)} - (\alpha_n^{(k)})^2)\right) \sum_{s,a} \Delta_k(s,a)^2 + (\alpha_n^{(k)})^2 \left(2\sigma^2 + 2\gamma^2 \|Q_n^{(0)}\|_\infty^2\right) \\
&= \left(1 - \xi(2\alpha_n^{(k)} - (\alpha_n^{(k)})^2)\right) E_n^{(k)} + (\alpha_n^{(k)})^2 C_n,
\end{aligned}$$

where in the penultimate line we used $\sum_{s,a} \pi_{n,k}(s,a)\Delta_k(s,a)^2 \geq \xi \sum_{s,a} \Delta_k(s,a)^2$. Moreover, we defined

$$C_n := 2\sigma^2 + 4\gamma^2(\|Q_n^{(0)} - Q^*\|_\infty^2 + \|Q^*\|_\infty^2),$$

which satisfies

$$2\sigma^2 + 2\gamma^2 \|Q_n^{(0)}\|_\infty^2 \leq C_n,$$

since $\|Q_n^{(0)}\|_\infty^2 \leq 2(\|Q_n^{(0)} - Q^*\|_\infty^2 + \|Q^*\|_\infty^2)$. Note that $Q_n^{(0)}$, $\|Q_n^{(0)}\|_\infty$, and hence $C_n$, are $\mathcal{F}_{n,0}$-measurable. Applying the tower property, we obtain the same scalar recursion but conditioned on $\mathcal{F}_{n,0}$:

$$\mathbb{E}\left[\|Q_n^{(k+1)} - T^* Q_n^{(0)}\|_2^2 \,\Big|\, \mathcal{F}_{n,0}\right] \leq \left(1 - \xi(2\alpha_n^{(k)} - (\alpha_n^{(k)})^2)\right) \mathbb{E}\left[\|Q_n^{(k)} - T^* Q_n^{(0)}\|_2^2 \,\Big|\, \mathcal{F}_{n,0}\right] + (\alpha_n^{(k)})^2 C_n.$$

To simplify the bound, note that $2\alpha_n^{(k)} - (\alpha_n^{(k)})^2 = \alpha_n^{(k)} + (\alpha_n^{(k)} - (\alpha_n^{(k)})^2)$, where $(\alpha_n^{(k)} - (\alpha_n^{(k)})^2) \geq 0$ since $0 < \alpha_n^{(k)} \leq 1$. Thus, we can write

$$1 - \xi(2\alpha_n^{(k)} - (\alpha_n^{(k)})^2) = 1 - \xi\alpha_n^{(k)} - \xi(\alpha_n^{(k)} - (\alpha_n^{(k)})^2)$$

implying

$$1 - \xi(2\alpha_n^{(k)} - (\alpha_n^{(k)})^2) \leq 1 - \xi\alpha_n^{(k)}.$$

Using this bound, the recursion simplifies to

$$\mathbb{E}\left[\|Q_n^{(k+1)} - T^* Q_n^{(0)}\|_2^2 \,\Big|\, \mathcal{F}_{n,0}\right] \leq \left(1 - \xi\alpha_n^{(k)}\right) \mathbb{E}\left[\|Q_n^{(k)} - T^* Q_n^{(0)}\|_2^2 \,\Big|\, \mathcal{F}_{n,0}\right] + (\alpha_n^{(k)})^2 C_n.$$

$\square$

*Proof of Theorem 4.5.* Let $e_n^{(k)} := \mathbb{E}[\|Q_n^{(k)} - T^* Q_n^{(0)}\|_2^2 \mid \mathcal{F}_{n,0}]$. Firstly, we observe that by definition $\alpha_n^{(0)} = 1$, such that $\alpha_n^{(k)} \leq 1$ for all $k \geq 0$. Thus, we can apply Proposition B.1 to obtain the recursion

$$e_n^{(k+1)} \leq (1 - \alpha_n^{(k)}\xi)e_n^{(k)} + (\alpha_n^{(k)})^2 C_n, \quad k \geq 0,$$

where $\alpha_n^{(k)} = \frac{2}{\xi(k + \frac{2}{\xi})}$. We will now show for $k \geq 1$ that $e_n^{(k)} \leq \frac{\Gamma}{k + \frac{2}{\xi}}$, where

$$\Gamma := \left(\frac{2}{\xi} + 1\right) \max(e_n^{(0)}, \tfrac{2C_n}{\xi}).$$

For $k = 1$, we have

$$e_n^{(1)} \leq (1 - \alpha_n^{(0)}\xi)e_n^{(0)} + (\alpha_n^{(0)})^2 C_n = (1 - \xi)e_n^{(0)} + C_n \leq \left(1 - \frac{\xi}{2}\right)\frac{\Gamma}{\frac{2}{\xi} + 1} \leq \frac{\Gamma}{\frac{2}{\xi} + 1},$$

where we used that $\xi \in (0, 1]$.

Now, suppose that the upper bound $e_n^{(k)} \leq \frac{\Gamma}{k+\frac{2}{\xi}}$ is satisfied for some $k \geq 1$. Using the basic expansion

$$\frac{\Gamma}{k+\frac{2}{\xi}+1} = \frac{\Gamma}{k+\frac{2}{\xi}} - \frac{\Gamma}{(k+\frac{2}{\xi})(k+\frac{2}{\xi}+1)}$$

it follows that

$$
\begin{aligned}
e_n^{(k+1)} &\leq (1 - \alpha_n^{(k)}\xi)e_n^{(k)} + (\alpha_n^{(k)})^2 C_n \leq \left(1 - \frac{2}{k+\frac{2}{\xi}}\right)\frac{\Gamma}{k+\frac{2}{\xi}} + \frac{4C_n}{\xi^2}\frac{1}{(k+\frac{2}{\xi})^2} \\
&= \frac{\Gamma}{k+1+\frac{2}{\xi}} + \frac{\Gamma}{(k+\frac{2}{\xi})(k+1+\frac{2}{\xi})} - \frac{2\Gamma}{(k+\frac{2}{\xi})^2} + \frac{4C_n}{\xi^2}\frac{1}{(k+\frac{2}{\xi})^2} \\
&= \frac{\Gamma}{k+1+\frac{2}{\xi}} + \frac{\frac{k+\frac{2}{\xi}}{k+\frac{2}{\xi}+1}\Gamma - 2\Gamma + \frac{4C_n}{\xi^2}}{(k+\frac{2}{\xi})^2} \\
&\leq \frac{\Gamma}{k+1+\frac{2}{\xi}} + \frac{-\Gamma + \frac{4C_n}{\xi^2}}{(k+\frac{2}{\xi})^2} \\
&\leq \frac{\Gamma}{k+1+\frac{2}{\xi}} + \frac{\frac{4C_n}{\xi^2} - \Gamma}{(k+\frac{2}{\xi})^2} \\
&\leq \frac{\Gamma}{k+1+\frac{2}{\xi}} .
\end{aligned}
$$

Here, we have used that

$$\Gamma = \left(\frac{2}{\xi}+1\right)\max\left(e_n^{(0)}, \frac{4C_n}{\frac{2}{\xi}\xi^2}\right) \geq \left(\frac{2}{\xi}+1\right)\frac{4C_n}{\frac{2}{\xi}\xi^2} \geq \frac{4C_n}{\xi^2},$$

as $\frac{2}{\xi} \geq 1$, which ensures $\frac{4C_n}{\xi^2} - \Gamma \leq 0$. We can directly apply the induction result to obtain

$$e_n^{(k)} \leq \frac{(\frac{2}{\xi}+1)\max(e_n^{(0)}, \frac{2C_n}{\xi})}{k+\frac{2}{\xi}} \leq \frac{(\frac{2}{\xi}+1)(e_n^{(0)} + \frac{2C_n}{\xi})}{k+\frac{2}{\xi}}, \quad \text{for all } k \geq 1.$$

Recall that

$$C_n = 2\sigma^2 + 4\gamma^2\left(\|Q_n^{(0)} - Q^*\|_\infty^2 + \|Q^*\|_\infty^2\right).$$

We can upper bound each initial SGD error via

$$
\begin{aligned}
e_n^{(0)} &= \|Q_n^{(0)} - Q^* + T^*Q^* - T^*Q_n^{(0)}\|_2^2 \\
&\leq |\mathcal{S}||\mathcal{A}| \cdot \|Q_n^{(0)} - Q^* + T^*Q^* - T^*Q_n^{(0)}\|_\infty^2 \\
&\leq |\mathcal{S}||\mathcal{A}| \cdot \left(\|Q_n^{(0)} - Q^*\|_\infty + \|T^*Q^* - T^*Q_n^{(0)}\|_\infty\right)^2 \\
&\leq |\mathcal{S}||\mathcal{A}|(1+\gamma)^2\|Q_n^{(0)} - Q^*\|_\infty^2,
\end{aligned}
$$

where we use the triangle inequality and the $\gamma$-contraction property of $T^*$.

Finally, we set

$$c_1 := \left(\frac{2}{\xi}+1\right)|\mathcal{S}||\mathcal{A}|(1+\gamma)^2 + \left(\frac{16}{\xi^2}+\frac{8}{\xi}\right)\gamma^2, \qquad c_2 := \left(\frac{8}{\xi^2}+\frac{4}{\xi}\right)\left(\sigma^2 + 2\gamma^2\|Q^*\|_\infty^2\right)$$

to deduce

$$e_n^{(k+1)} \leq \frac{c_1\|Q_n^{(0)} - Q^*\|_\infty^2 + c_2}{k+\frac{2}{\xi}} .$$

This completes the proof. □

## B.2. Convergence Result

*Proof of Corollary 4.6.* Let $\mathcal{F} := (\mathcal{F}_{n,0})_{n \in \mathbb{N}_0}$ be the filtration at inner loop starts. Using the bound $\|x\|_\infty \leq \|x\|_2$ together with Jensen's inequality, it holds

$$\mathbb{E}\Big[\|Q_n^{(K_n)} - T^* Q_n^{(0)}\|_\infty \,\Big|\, \mathcal{F}_{n,0}\Big] \leq \Big(\mathbb{E}\Big[\|Q_n^{(K_n)} - T^* Q_n^{(0)}\|_2^2 \,\Big|\, \mathcal{F}_{n,0}\Big]\Big)^{1/2}.$$

Plugging in the inner loop bound of Theorem 4.5 at the final iterate $k = K_n$ yields

$$\mathbb{E}\Big[\|Q_n^{(K_n)} - T^* Q_n^{(0)}\|_2^2 \,\Big|\, \mathcal{F}_{n,0}\Big] \leq \frac{c_1 \|Q_n^{(0)} - Q^*\|_\infty^2 + c_2}{K_n + \frac{2}{\xi}} \leq \frac{c_1 E_n^2 + c_2}{K_n}.$$

Taking square roots, applying subadditivity and adding the $\gamma E_n$ term, which is $\mathcal{F}_{n,0}$-measurable, gives

$$\mathbb{E}[E_{n+1} \,|\, \mathcal{F}_{n,0}] \leq \gamma E_n + \frac{\sqrt{c_1} E_n + \sqrt{c_2}}{\sqrt{K_n}}. \tag{18}$$

Hence, from Proposition 4.1 we have

$$\mathbb{E}\Big[\|Q_{n+1}^{(0)} - Q^*\|_\infty \,\Big|\, \mathcal{F}_{n,0}\Big] \leq \gamma \|Q_n^{(0)} - Q^*\|_\infty + \frac{\sqrt{c_1}\|Q_n^{(0)} - Q^*\|_\infty + \sqrt{c_2}}{\sqrt{K_n}},$$

for all $n \geq 0$ and the constants $c_1$, $c_2$ defined in Theorem 4.5. Set $Z_n := \|Q_n^{(0)} - Q^*\|_\infty$ and define deterministic, nonnegative sequences

$$A_n := \frac{\sqrt{c_1}}{\sqrt{K_n}}, \qquad B_n := \frac{\sqrt{c_2}}{\sqrt{K_n}}, \qquad D_n := 1 - \gamma \in (0,1).$$

Then the one–step inequality rewrites as

$$\mathbb{E}[Z_{n+1} \,|\, \mathcal{F}_{n,0}] \leq Z_n (1 + A_n - D_n) + B_n.$$

Suppose that $\sum_{n=0}^\infty K_n^{-\frac{1}{2}} < \infty$. Then we have

$$\sum_{n=1}^\infty A_n = \sum_{n=1}^\infty \frac{\sqrt{c_1}}{\sqrt{K_n}} < \infty,$$

and similarly $\sum_{n=1}^\infty B_n < \infty$, while

$$\sum_{n=1}^\infty D_n = \sum_{n=1}^\infty (1 - \gamma) = \infty.$$

Therefore, the conditions of Corollary 4.2 are satisfied for the adapted, nonnegative process $(Z_n)_{n \geq 0}$. Hence $Z_n \to 0$ almost surely, i.e., we have verified that

$$\|Q_n^{(0)} - Q^*\|_\infty \longrightarrow 0 \qquad \text{a.s. as } n \to \infty.$$

$\square$

## B.3. Auxiliary Result

**Theorem B.2** (Theorem 1 in (Robbins & Siegmund, 1971))**.** *Let $(\Omega, \mathcal{A}, \mathcal{F}, \mathbb{P})$ be a filtered probability space, $(Z_k)_{k \in \mathbb{N}}$, $(A_k)_{k \in \mathbb{N}}$, $(B_k)_{k \in \mathbb{N}}$ and $(C_k)_{k \in \mathbb{N}}$ be non-negative and $\mathcal{F}$-adapted stochastic processes such that*

$$\sum_{k=0}^\infty A_k < \infty \quad \text{and} \quad \sum_{k=0}^\infty B_k < \infty$$

*almost surely. Moreover, suppose*

$$\mathbb{E}[Z_{k+1} \,|\, \mathcal{F}_k] \leq Z_k(1 + A_k) + B_k - C_k.$$

*Then*

1. *there exists an almost surely finite random variable $Z_\infty$ such that $Z_k \to Z_\infty$ almost surely for $k \to \infty$,*

2. *it holds that $\sum_{k=0}^\infty C_k < \infty$ almost surely.*

## C. TUF Design: Details and Proofs of Section 5

### C.1. Quasi-Optimal TUF Schedules

In this section, we give more details on the construction of the optimal TUF design. Recall that we want to choose the cycle lengths $(K_n)$ with the aim of minimizing the total computational cost for a given target accuracy. The cost of computing $Q_n^{(K_n)}$ is given as

$$\text{cost}(Q_n^{(K_n)}) = \sum_{j=1}^{n} K_j \,.$$

Given $\varepsilon > 0$, the goal is to select $N$ and $(K_n)_{n=0}^{N-1}$ that solve the (relaxed) constrained optimization problem

$$\min_{N \in \mathbb{N}, (K_n)_{n=0}^{N-1} \in \mathbb{R}^N} \text{cost}(Q_{N-1}^{(K_N)}) \quad \text{s.t.} \quad e_N := \mathbb{E}[E_N] \leq \varepsilon \,. \tag{19}$$

One can define an effective contraction factor $\mu \in (\gamma, 1)$ and enforce the minimal cycle lengths to be sufficiently large such that for all $n \in \mathbb{N}_0$,

$$0 < \gamma + \sqrt{\frac{c_1}{K_n}} \leq \mu < 1 \,. \tag{20}$$

More precisely, we require $\min_n K_n \geq K_{\min}(\mu) := \frac{c_1}{(\mu-\gamma)^2}$. Thus, taking expectation on both sides in Equation (11) and iterating over $n \in \mathbb{N}$ yields

$$e_{n+1} \leq \mu \, e_n + \sqrt{\frac{c_2}{K_n}}, \qquad n \in \mathbb{N}_0 \,.$$

Unrolling this recursion gives $e_n \leq \mu^n e_0 + \sum_{j=0}^{n-1} \mu^{n-1-j} \sqrt{\frac{c_2}{K_j}}$ for all $n \in \mathbb{N}$. From now on, we optimize (19) with

$$e_n := e_N(K_j : j = 0, \ldots, N-1) := \mu^n e_0 + \sum_{j=0}^{n-1} \mu^{n-1-j} \sqrt{\frac{c_2}{K_j}}, \quad n \in \mathbb{N} \tag{21}$$

under the additional constraint that $\min_n K_n \geq K_{\min}(\mu) := \frac{c_1}{(\mu-\gamma)^2}$. For simplicity, we allow the cycle lengths $(K_n)$ to be positive real numbers. If we consider a fixed cycle length we simply write $e_N = e_N(K)$. As discussed in Section 5, we enforce the error constraint $e_N \leq \varepsilon$ in (19) by decomposing it into

$$\mu^N e_0 \leq \frac{\varepsilon}{2} \quad \text{and} \quad \sum_{j=0}^{N-1} \mu^{N-1-j} \sqrt{\frac{c_2}{K_j}} \leq \frac{\varepsilon}{2} \,.$$

This has the advantage that it fixes the number of required outer loops to $N(\varepsilon) \geq \left\lceil \frac{\log(\varepsilon/(2e_0))}{\log(\mu)} \right\rceil$ and reduces the design problem to selecting $K_n(\varepsilon)$, $n = 0, \ldots, N(\varepsilon) - 1$. While this results in a simpler constraint optimization problem to be solved, a-priori, it is unclear whether an asymmetric splitting of the two error terms may improve the complexity. Therefore, we first introduce our notion of optimality. We are interested in TUF design choices that, asymptotically for $\varepsilon \to 0$, can only be improved by constants.

**Definition C.1** (Quasi-optimal TUF schedule). For $N \in \mathbb{N}$ and cycle lengths $(K_j, j = 0, \ldots, N-1) \subset \mathbb{R}_+$ consider $e_N = e_N(K_j : j = 0, \ldots, N-1)$ defined in (19).

1. Let $(N(\varepsilon), K(\varepsilon, N(\varepsilon)))_{\varepsilon>0}$ be a family of integers $N(\varepsilon) \in \mathbb{N}$ and reals $K(\varepsilon, N(\varepsilon)) \in \mathbb{R}_+$ that satisfy $e_{N(\varepsilon)}(K(\varepsilon, N(\varepsilon))) \leq \varepsilon$ for all sufficiently small $\varepsilon > 0$. We say that $(N(\varepsilon), K(\varepsilon, N(\varepsilon)))_{\varepsilon>0}$ is a *quasi-optimal fixed target update frequency schedule* if

$$N(\varepsilon) \, K(\varepsilon, N(\varepsilon)) \in \mathcal{O}\big( \inf \big\{ \hat{N}\hat{K} : e_{\hat{N}}(\hat{K}) \leq \varepsilon, \, \hat{N} \in \mathbb{N}, \, \hat{K} > 0 \big\} \big) \quad \text{as } \varepsilon \to 0 \,.$$

2. Let $(N(\varepsilon), K_j(\varepsilon, N(\varepsilon)), 0 \leq j \leq N(\varepsilon) - 1)_{\varepsilon>0}$ be a family of integers $N(\varepsilon) \in \mathbb{N}$ and sequences $(K_j(\varepsilon, N(\varepsilon)), j = 0, \ldots, N(\varepsilon) - 1) \subset \mathbb{R}_+$ such that $e_{N(\varepsilon)}(K_j(\varepsilon, N(\varepsilon)), j = 0, \ldots, N(\varepsilon) - 1) \leq \varepsilon$ for all sufficiently small $\varepsilon > 0$. We say that $(N(\varepsilon), K_j(\varepsilon, N(\varepsilon)), 0 \leq j \leq N(\varepsilon) - 1)_{\varepsilon>0}$ is a *quasi-optimal variable target update frequency schedule* if

$$\sum_{j=0}^{N(\varepsilon)-1} K_j(\varepsilon, N(\varepsilon)) \in \mathcal{O}\big( \inf \big\{ \sum_{j=0}^{\hat{N}-1} \hat{K}_j : e_{\hat{N}}(\hat{K}_j, j = 0, \ldots, \hat{N}-1) \leq \varepsilon, \, \hat{N} \in \mathbb{N}, \, \hat{K}_j > 0 \, \forall j < \hat{N}-1 \big\} \big) \quad \text{as } \varepsilon \to 0 \,.$$

### C.2. Proofs of Sample Complexities

*Proof of Theorem 5.1.* First, we suppress the lower bound $K \geq K_{\min}$ required in (20). (21) with constant cycle length $K$ gives

$$e_n = \mu^n e_0 + \frac{\sqrt{c_2}}{\sqrt{K}} \frac{1 - \mu^n}{1 - \mu}.$$

We consider the separate the constraints

$$\mu^n e_0 \leq \tfrac{\varepsilon}{2} \quad \text{and} \quad \frac{\sqrt{c_2}}{\sqrt{K}} \frac{1 - \mu^n}{1 - \mu} \leq \frac{\varepsilon}{2}. \tag{22}$$

Similarly to Theorem 5 in (Weissmann et al., 2022), solving these constraints with equality and for $n$ and $K$ gives the choices:

$$N(\varepsilon) = \left\lceil \frac{\log(\varepsilon/(2e_0))}{\log(\mu)} \right\rceil, \quad K(\varepsilon) = \frac{4c_2}{\varepsilon^2} \cdot \left( \frac{1 - \mu^{N(\varepsilon)}}{1 - \mu} \right)^2$$

Next we set $\mu = \frac{1+\gamma}{2} \in (\gamma, 1)$ and assume $\varepsilon \leq (1 - \gamma)\sqrt{\frac{c_2}{c_1}}$ such that

$$\frac{c_1}{(\mu - \gamma)^2} = \frac{4c_1}{(1 - \gamma)^2} \leq \frac{4c_2(1 - \gamma)^2}{\varepsilon^2(1 - \mu)^2} = \frac{16c_2}{\varepsilon^2}$$

where we have used $\mu - \gamma = \frac{1}{2}(1 - \gamma) = 1 - \mu$. Therefore, we have

$$K(\varepsilon) \geq K_{\min}(\mu) = \frac{c_1}{(\mu - \gamma)^2}$$

and associated computational cost

$$\text{cost}(Q_{N(\varepsilon)}^{(0)}) = N(\varepsilon)\, K \leq \left\lceil \frac{\log((2e_0)/\varepsilon)}{\log(\mu^{-1})} \right\rceil \frac{4c_2}{\varepsilon^2} \left( \frac{1}{1 - \mu} \right)^2$$

$$\leq \left\lceil \frac{\log((2e_0)/\varepsilon)}{\log(2(1 + \gamma)^{-1})} \right\rceil \frac{16c_2}{\varepsilon^2} \left( \frac{1}{1 - \gamma} \right)^2.$$

If we additionally assume $R(s, a) \in [-1, 1]$ it follows directly that $\sigma^2 \leq 1$, $\|Q^*\|_\infty \leq \frac{1}{1-\gamma}$, and $e_0 \leq \frac{2}{1-\gamma}$. which implies

$$c_2 \lesssim \frac{1}{\xi^2}\left(1 + \frac{\gamma^2}{(1 - \gamma^2)^2}\right) = \frac{1}{\xi^2}\left(\frac{1 - 2\gamma + 2\gamma^2}{(1 - \gamma)^2}\right) \lesssim \frac{1}{\xi^2(1 - \gamma)^2}. \tag{23}$$

The resulting sample complexity is of order

$$\text{cost}(Q_{N(\varepsilon)}^{(0)}) = \mathcal{O}\left( \frac{1}{(1 - \gamma)^4 \xi^2 \log(2(1 + \gamma)^{-1})}\, \varepsilon^{-2}\, \log\frac{4}{(1 - \gamma)\varepsilon} \right). \tag{24}$$

Finally, we have to justify the quasi-optimality. Suppose we select

$$\mu^n e_0 \leq \lambda\varepsilon \quad \text{and} \quad \frac{\sqrt{c_2}}{\sqrt{K}} \frac{1 - \mu^n}{1 - \mu} \leq (1 - \lambda)\varepsilon.$$

for arbitrary $\lambda \in (0, 1)$ instead of the symmetric decomposition with $\lambda = 1/2$ in (22). Then the optimal choices would result in $\hat{N}(\varepsilon) = N(2\lambda\varepsilon) \geq N(2\varepsilon)$ and $\hat{K}(\varepsilon) = K(2\lambda\varepsilon) \geq K(2\varepsilon)$ leading to computational cost lower bounded by

$$\hat{N}(\varepsilon)\hat{K}(\varepsilon) \geq \left\lceil \frac{\log(e_0/\varepsilon)}{\log(2(1 + \gamma)^{-1})} \right\rceil \frac{4c_2}{\varepsilon^2} \left( \frac{1}{1 - \gamma} \right)^2.$$

More precisely, we have that

$$\left\lceil \frac{\log(e_0/\varepsilon)}{\log(2(1 + \gamma)^{-1})} \right\rceil \frac{4c_2}{\varepsilon^2} \left( \frac{1}{1 - \gamma} \right)^2 \leq \inf\left\{ \hat{N}\hat{K} : e_{\hat{N}}(\hat{K}) \leq \varepsilon,\ \hat{N} \in \mathbb{N},\ \hat{K} > 0 \right\} \leq \left\lceil \frac{\log((2e_0)/\varepsilon)}{\log(2(1 + \gamma)^{-1})} \right\rceil \frac{16c_2}{\varepsilon^2} \left( \frac{1}{1 - \gamma} \right)^2$$

for all $0 < \varepsilon \leq (1 - \gamma)\sqrt{\frac{c_2}{c_1}}$. □

*Proof of Theorem 5.3.* Since we do not fix the cycle lengths, we begin with the error decomposition of (21) into

$$\mu^N e_0 \le \frac{\varepsilon}{2} \quad \text{and} \quad \sum_{j=0}^{N-1} \mu^{N-1-j} \sqrt{\frac{c_2}{K_j}} \le \frac{\varepsilon}{2}.$$

Ignoring the required minimal cycle lengths, the application of Theorem 8 in (Weissmann et al., 2022) with the identifications

$$c = \mu, \qquad \ell_j = \frac{K_j}{c_2}, \qquad \alpha = \tfrac{1}{2}$$

gives a solution of the relaxed constrained problem

$$\min_{(K_n)_{n=0}^{N(\varepsilon)-1}} \mathrm{cost}(Q_{N(\varepsilon)}^{(0)}) \quad \text{s.t.} \quad \sum_{j=0}^{N(\varepsilon)-1} \mu^{N(\varepsilon)-1-j} \sqrt{\frac{c_2}{K_j}} \le \frac{\varepsilon}{2}.$$

This solution can be written as

$$N(\varepsilon) = \left\lceil \frac{\log(\frac{\varepsilon}{2e_0})}{\log(\mu)} \right\rceil$$

and

$$K_j(\varepsilon) = C_\varepsilon \mu^{\frac{2}{3}(N(\varepsilon)-1-j)} \quad \text{with} \quad C_\varepsilon = \frac{4c_2}{\varepsilon^2} \left( \frac{1 - \mu^{\frac{2}{3}N(\varepsilon)}}{1 - \mu^{\frac{2}{3}}} \right)^2.$$

Next, we seek a (sufficient) condition on $\varepsilon > 0$ for which

$$K_j(\varepsilon) \ge K_0(\varepsilon) > \frac{4c_2}{(1-\gamma)^2}.$$

Substituting the definitions and canceling $4c_2$ yields

$$\frac{1}{\varepsilon^2} \left( \frac{1 - \mu^{\frac{2}{3}N(\varepsilon)}}{1 - \mu^{\frac{2}{3}}} \right)^2 \mu^{\frac{2}{3}(N(\varepsilon)-1)} > \frac{1}{(1-\gamma)^2}.$$

Taking square-roots (all quantities are nonnegative) gives the equivalent inequality

$$\varepsilon < (1-\gamma) \, \mu^{\frac{N(\varepsilon)-1}{3}} \, \frac{1 - \mu^{\frac{2}{3}N(\varepsilon)}}{1 - \mu^{\frac{2}{3}}}. \tag{25}$$

From the ceiling definition of $N(\varepsilon)$ we have the bracketing

$$\mu^{N(\varepsilon)} \le \frac{\varepsilon}{2e_0} < \mu^{N(\varepsilon)-1}.$$

Since $\mu \in (0, 1)$, this implies

$$\mu^{\frac{N(\varepsilon)-1}{3}} > \left( \frac{\varepsilon}{2e_0} \right)^{1/3}, \qquad 1 - \mu^{\frac{2}{3}N(\varepsilon)} \ge 1 - \left( \frac{\varepsilon}{2e_0} \right)^{2/3}.$$

Inserting these bounds into (25) yields the sufficient condition

$$\varepsilon < (1-\gamma) \left( \frac{\varepsilon}{2e_0} \right)^{1/3} \frac{1 - \left( \frac{\varepsilon}{2e_0} \right)^{2/3}}{1 - \mu^{2/3}}. \tag{26}$$

Let $x := \left( \frac{\varepsilon}{2e_0} \right)^{1/3} > 0$. Then $\varepsilon = 2e_0 x^3$ and (26) becomes

$$2e_0 x^3 < (1-\gamma) \, x \, \frac{1 - x^2}{1 - \mu^{2/3}}.$$

Dividing by $x$ and setting $\nu := \frac{1-\gamma}{1-\mu^{2/3}} > 0$ gives

$$2e_0 x^2 < \nu(1 - x^2) \iff (2e_0 + \nu)x^2 < \nu \iff x^2 < \frac{\nu}{2e_0 + \nu}.$$

Returning to $\varepsilon$ yields the explicit sufficient bound

$$\varepsilon < 2e_0 \left( \frac{\nu}{2e_0 + \nu} \right)^{3/2}$$

which guarantees $K_0(\varepsilon) > \frac{4c_2}{(1-\gamma)^2}$.

The total costs can be computed by applying the geometric series as

$$\text{cost}(Q_{N(\varepsilon)}^{(0)}) = \sum_{j=0}^{N(\varepsilon)-1} \max(C_\varepsilon \mu^{\frac{2}{3}(N(\varepsilon)-1-j)}, K_{\min}) = \left( \frac{\varepsilon}{2\sqrt{c_2}} \right)^{-2} \left( \frac{1 - \mu^{\frac{2}{3}N(\varepsilon)}}{1 - \mu^{2/3}} \right)^3.$$

Further, we use $\mu^{N(\varepsilon)} \le \varepsilon/(2e_0)$, such that $\mu^{\frac{2}{3}N(\varepsilon)} \to 0$ as $\varepsilon \to 0$, together with $1 - \mu = \frac{1}{2}(1 - \gamma)$, such that

$$\left( \frac{1 - \mu^{\frac{2}{3}N(\varepsilon)}}{1 - \mu^{2/3}} \right)^3 = \mathcal{O}\Big( \frac{1}{(1 - \mu^{2/3})^3} \Big) = \mathcal{O}\Big( \frac{1}{(1 - \mu)^3} \Big) = \mathcal{O}\Big( \frac{1}{(1 - \gamma)^3} \Big),$$

where the second last step uses $1 - \mu^{2/3} \ge \frac{2}{3}(1 - \mu)$, due to concavity of $x \mapsto x^{2/3}$ on $[0,1]$. Together with $c_2 \lesssim \frac{1}{\xi^2}\big(\frac{1-2\gamma+2\gamma^2}{(1-\gamma)^2}\big) \lesssim \frac{1}{\xi^2(1-\gamma)^2}$ from Eq. (23), we have

$$\text{cost}(Q_{N(\varepsilon)}^{(0)}) = \mathcal{O}\Big( \frac{1}{\xi^2(1 - \gamma)^5 \varepsilon^2} \Big). \tag{27}$$

The argument for the quasi-optimality of the derived frequency update follows in line with the argument of the proof of Theorem 5.1 showing that

$$\sum_{j=0}^{N(2\varepsilon)-1} K_j(2\varepsilon) = \left( \frac{\varepsilon}{\sqrt{c_2}} \right)^{-2} \left( \frac{1 - \gamma^{\frac{2}{3}N(2\varepsilon)}}{1 - \gamma^{2/3}} \right)^3$$

$$\le \inf \Big\{ \sum_{j=0}^{\hat{N}-1} \hat{K}_j : e_{\hat{N}}(\hat{K}_j, \, j = 0, \ldots, \hat{N}-1) \le \varepsilon, \, \hat{N} \in \mathbb{N}, \, \hat{K}_j > 0 \, \forall j < \hat{N}-1 \Big\}$$

$$\le \sum_{j=0}^{N(\varepsilon)-1} K_j(\varepsilon) = \left( \frac{\varepsilon}{2\sqrt{c_2}} \right)^{-2} \left( \frac{1 - \gamma^{\frac{2}{3}N(\varepsilon)}}{1 - \gamma^{2/3}} \right)^3.$$

$\square$

# D. Outlook

## D.1. Bridging Theory and Practice

In this section, we follow the suggestion of (Aitchison et al., 2023) and report preliminary results on the `NameThisGame` environment.

We begin by comparing the effect of the proposed increasing-cycle scheme for DQN between using SGD and ADAM to fit the networks to the target. In particular, in both cases we compare fixed TUFs with increasing cycles across a range of initial TUFs (ITUFs). Figure 5 shows the resulting performance averaged over 10 seeds. The top panel corresponds to SGD with a custom learning rate schedule that decreases linearly from 0.01 to 0.0001 within each target-freezing interval. The bottom panel reports results for ADAM with a constant learning rate of 0.0001. All other hyperparameters are set as the default choices made by Stable Baselines 3 (Raffin et al., 2021). Across both optimizers, higher constant TUFs yield faster initial

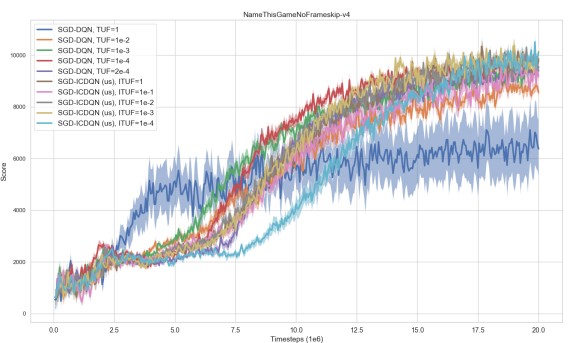 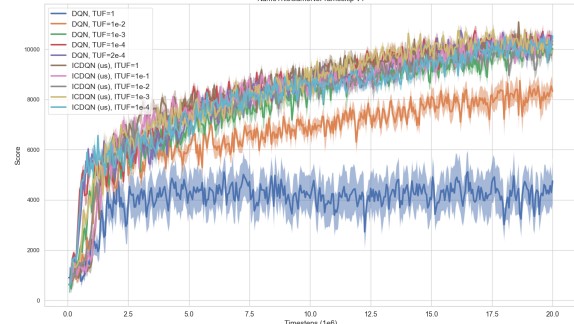

*Figure 5.* Score comparison of DQN with fixed and increasing cycles over different ITUFs on `NameThisGame`. Top: SGD with linearly decreasing learning rate per target-freezing interval. Bottom: ADAM with constant learning rate. Results are averaged over 10 seeds.

improvements but plateau at lower performance levels. In contrast, increasing cycles appear more stable with respect to the choice of ITUF and achieve performance comparable to well-tuned fixed schedules. Notably, this behavior seems to persist under ADAM despite the lack of theoretical guarantees in this setting.

To assess whether the observed effects extend beyond DQN, we additionally consider BTR + RISE (cf. (Clark et al., 2025b;a) and references therein), which is currently SOTA in terms of wall-clock time. Among other adaptations that do not change the fundamental nature of the update rule, this algorithm employs $N$-step bootstrapping, Double DQN, IQN, and Munchausen RL, which all retain the fundamental intuition of ADP aiming to use the contraction of the Bellman operator, making it plausible that our derived TUF schedule may still be beneficial. Figure 6 reports results averaged over 5 seeds for the recommended fixed and several increasing cycles across different ITUFs, where all other hyperparameters are set to the default values reported in (Clark et al., 2025b) and (Clark et al., 2025a). In this setting, increasing cycles exhibit improvements on longer time scales, with performance continuing to grow beyond the plateau observed for fixed schedules.

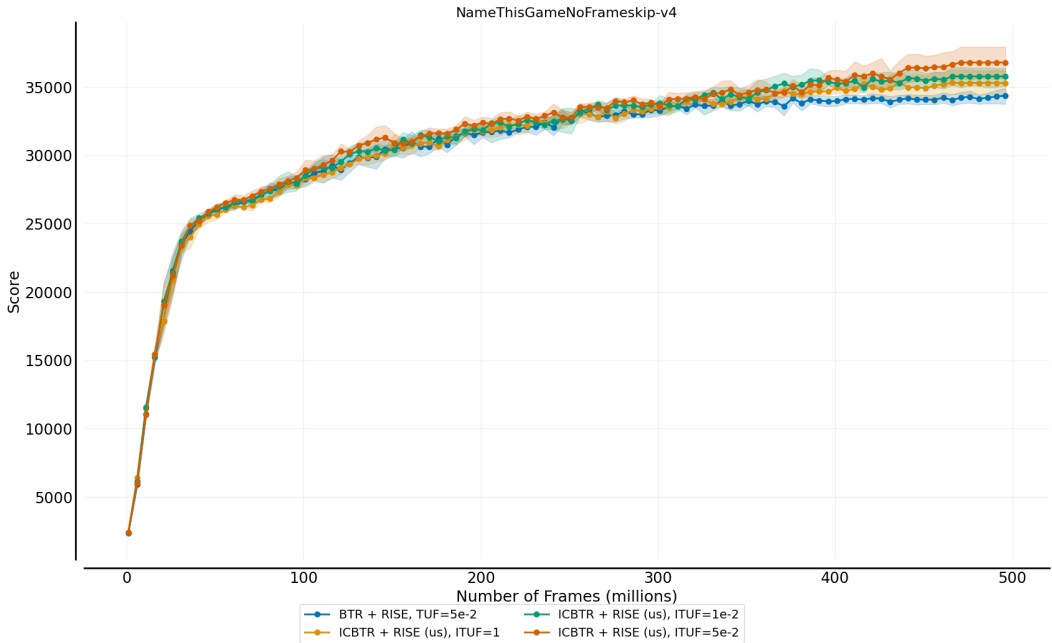

*Figure 6.* Score comparison of BTR + RISE for fixed and increasing cycles over different ITUFs on `NameThisGame`. Results are averaged over 5 seeds.

While preliminary, these findings suggest that the proposed scheme may transfer beyond the application of SGD to fit the target and to other ADP-style algorithms that rely on target network freezing as well.

## D.2. Accuracy-Triggered Target Updates

In the following, we describe details for an accuracy-triggered target update mentioned in Section 6. The approximate Bellman contraction perspective developed in this work suggests a natural extension beyond predetermined TUFs. Recall that the outer-loop dynamics satisfy

$$\mathbb{E}[E_{n+1} \mid \mathcal{F}_{n,0}] \le \gamma E_n + \eta_n,$$

where $\eta_n$ denotes the inner-loop approximation error in cycle $n$. This recursion highlights that outer-loop progress depends primarily on achieving a sufficiently accurate Bellman operator approximation, rather than on the number of inner iterations per se. Motivated by this observation, we consider *accuracy-triggered target updates*, in which the target network is synchronized once the inner-loop approximation error falls below a prescribed threshold. Instead of fixing the TUFs $1/K_n$ in advance, one specifies a sequence of accuracy levels $(\varepsilon_n)_{n \ge 1}$ and terminates the inner loop as soon as the corresponding accuracy criterion is met. This allows the computational effort per cycle to adapt to the local difficulty of approximating the Bellman operator.

**Online Estimation in the Tabular Setting.** In the tabular case, we outline a heuristic practical mechanism for estimating $\eta_n$ online during the inner loop. Recall that tabular Q-learning with frozen targets can be viewed as stochastic gradient descent on a quadratic Bellman regression objective. The temporal-difference error $\delta = r + \gamma \max_{a'} Q^-(s', a') - Q(s, a)$ constitutes a noisy sample of the gradient of the corresponding per-sample loss. For each state–action pair $(s, a)$, we maintain the empirical mean of observed TD-errors, updated online via standard averaging. We then aggregate these local statistics into a global stopping criterion

$$M = \frac{1}{|\mathcal{S}||\mathcal{A}|} \sum_{(s,a)} |\bar{\delta}(s, a)|,$$

which is small only if all state–action pairs exhibit consistently small average TD-errors. Since the tabular Bellman regression problems are strongly convex with unique minimizers characterized by vanishing gradients, the condition $M \approx 0$ indicates that the inner-loop optimization problems are approximately solved across the state–action space.

To ensure convergence, the approximation levels must be summable across cycles, as required by Proposition 4.1. Accordingly, a simple admissible choice is a polynomially decaying sequence $\varepsilon_n = n^{-p}$, $p > 1$, for which $\sum_{n \ge 1} \varepsilon_n < \infty$. In our implementation we use the concrete instance $\varepsilon_n = n^{-2}$.

Note that one may use alternative stopping rules, for instance based on relative changes in the value function instead of an absolute TD-error threshold.

---

**Algorithm 3** Periodic Q-Learning (Tabular) with Accuracy-Triggered Target Updates (ATQL)

---

1: **Input:** Maximum cycle length $K_{\max}$, minimum cycle length $K_{\min}$, target accuracies $\{\varepsilon_n\}_{n=1}^N$
2: Initialize $Q : \mathcal{S} \times \mathcal{A} \to \mathbb{R}$ arbitrarily, $Q^- = Q$
3: **for** Outer Bellman iteration with $n = 1$ **to** $N$ **do**
4:      Initialize counters $c(s, a) \leftarrow 0$ and means $\mu(s, a) \leftarrow 0$ for all $(s, a) \in \mathcal{S} \times \mathcal{A}$
5:      **for** Inner SGD iteration with $k = 1$ **to** $K_{\max}$ **do**
6:          Observe current state $s$, choose action $a$, e.g. $a \sim \varepsilon$-greedy($Q(\cdot, \cdot)$), execute action $a$, observe $r, s'$
7:          Compute TD error $\delta \leftarrow r + \gamma \max_{a' \in \mathcal{A}} Q^-(s', a') - Q(s, a)$
8:          Update: $Q(s, a) \leftarrow Q(s, a) + \alpha \delta$,   $c(s, a) \leftarrow c(s, a) + 1$
9:          $\mu(s, a) \leftarrow \mu(s, a) + \frac{\delta - \mu(s,a)}{c(s,a)}$,   $M \leftarrow \frac{1}{|\mathcal{S} \times \mathcal{A}|} \sum_{(\tilde{s}, \tilde{a})} |\mu(\tilde{s}, \tilde{a})|$
10:          **if** $k \ge K_{\min,n}$ **and** $M \le \varepsilon_n$ **then**
11:              **break**
12:          **end if**
13:      **end for**
14:      $Q^- \leftarrow Q$
15: **end for**

---

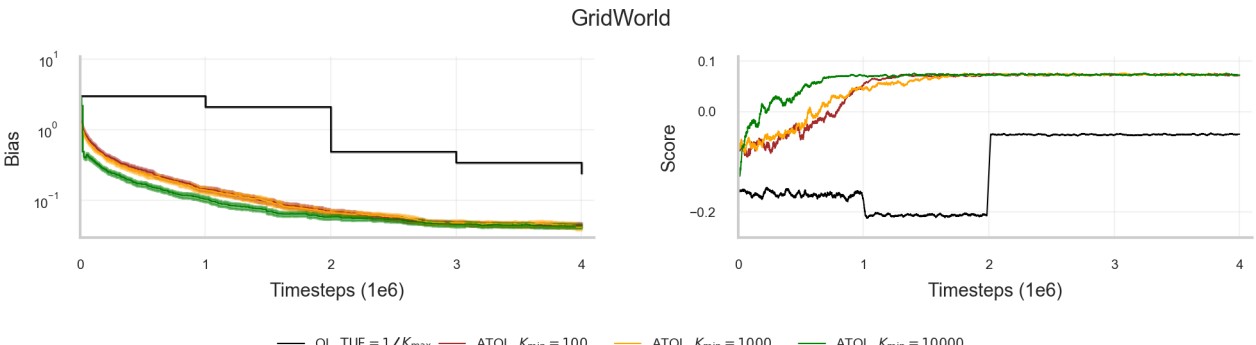

*Figure 7.* Q-learning with accuracy-triggered target updates, $K_{\min} \in \{100, 1000, 10000\}$ and $K_{\max} = 1e6$ on the GridWorld environment from Appendix A.2. Target accuracies are chosen by $\varepsilon_n = \frac{1}{n^2}$, motivated by the condition for convergence in Prop. 4.1. As comparison we include Q-learning with TUF $= 1/K_{\max} = 10^{-6}$ to illustrate the effect of accuracy-triggered target updates. Unnecessarily long inner loops for Bellman approximations are stopped early leading to an improved convergence behavior.

