# OpenReview forum: "The Role of Target Update Frequencies in Q-Learning"
_ICML.cc/2026/Conference — ICML 2026 regular_

### Official Review · Reviewer_TWsn · 2026-02-20

**Soundness:** 3
**Presentation:** 2
**Significance:** 3
**Originality:** 3
**Overall Recommendation:** 4
**Confidence:** 4

**Summary:**

This paper studies the effect of target update frequency in Q-Learning under tabular setting. In particular, the paper first consider Q-learning procedure as a bi-level optimization problem, where the inner loop approximates the target function and the outer loop conducts the value iteration. The paper first constructs the error upper bound of Bellman operator approximation with respect to $K_n$. Based on this result, the paper analyze two TUF schedule, namely constant and geometric schedule and provide the corresponding sample complexity.

**Compliance With Llm Reviewing Policy:**

Affirmed.

**Final Justification:**

This paper is generally well written and delivers a clear improvement over previous results. While the strong assumption undermines the strength of the paper, I think the overall contribution still merits a weak acceptance.

Rebuttal: The author promises a deeper discussion about the techniques, which should address most of my concerns.

**Key Questions For Authors:**

Questions:

1. Is it possible to extend the analysis to control high probability error?

2. Could the author provide more insight in why the analysis here remove a factor of $|S||A|$?

**Limitations:**

See weakness section.

**Strengths And Weaknesses:**

The strengths and weaknesses are listed as follows

Strengths:

1. The paper is clearly written, and the analysis is well organized and thus generally easy to follow.

2. The results are interesting, improving an $|S||A|$ upon previous results. The author also show that the logarithm dependency on $\epsilon$ can be removed via a geometric updating frequency schedule.

Weaknesses:

1. Strong Assumption: The analysis requires uniform state-action coverage. In MDP setting, the coverage depends on not only the policy but also the transition kernel. To ensure an uniform coverage, one generally require the simulator to be able to start from arbitrary $(s,a)$, which might not be feasible in some application.

2. Lack of High Level Illustration: The introduction to the main results lacks corresponding intuitive explanation. For example, the introduction to $\mu$ in (13) is abrupt. Around Theorem 5.3, there is no explanation why picking geometrically decaying schedule removes the logarithm factor. Lacking of such illustration make the paper looks like a technical report.

3. (Minor) Writing: Some parentheses are not matched, e.g., line 152, 159, 836. At line 800, $R$ is not defined.

---

> ### Author Rebuttal · Authors · 2026-03-31
>
> We thank the reviewer for reviewing our paper and overall positive feedback. Below we provide detailed responses to the raised comments and questions.
>
> ## Weaknesses
> **Weakness 1:** We agree that the considered coverage assumptions are strong. Our goal in this work, however, is to provide the first principled characterization of how the target update frequency should be chosen, and to do so in a setting where the effect of delayed targets can be isolated cleanly. In particular, our analysis identifies the bias–variance trade-off induced by frozen targets and establishes convergence guarantees without the additional $\log(\epsilon^{-1})$ factor that appears in most Q-learning theory papers. Future work will include extensions to $\epsilon$-greedy asynchronous dynamics in $N$-ergodic (meaning we can reach each state in at most $N$ steps) MDPs, where we expect the inner loop to scale with $\lfloor\frac{k}{N}\rfloor$ and $\epsilon^{\lfloor\frac{k}{N}\rfloor}$ (with $\epsilon$ from $\epsilon$-greedy) instead of $k$ and $\xi$, yielding the same qualitative behavior and meaning the optimal TUFs change only in a constant.
>
> **Weakness 2:** Thank you very much for this comment. Regarding the introduction of $\mu$, we added the following paragraph
>
> In view of (11), one has $e_{n+1}\le \Big( \gamma + \sqrt{\frac{c_1}{K_n}} \Big) e_n + \sqrt{\frac{c_2}{K_n}}$. Thus, for every effective contraction factor \(\mu\in(\gamma,1)\) we can choose sufficiently large TUFs $(K_n)$ such that $\min_{n} K_n \ge K_{\min}(\mu) := \frac{c_1}{(\mu-\gamma)^2}$ so that
> $$
>     e_{n+1} \le  \mu e_n + \sqrt{\frac{c_2}{K_n}}
> $$
> and, iteratively,
> $$
>     e_n \le \mu^n e_0 + \sum_{j=0}^{n-1}\mu^{n-1-j}\sqrt{\frac{c_2}{K_j}}, \quad n \in \mathbb N.
> $$
>
> Regarding an intuitive explanation why geometrically increasing: The key point is that the final error bound after $N:=N(\varepsilon)$ target updates has the form
> $$e_N \le \mu^N e_0 + \sum_{j=0}^{N-1}\mu^{N-1-j}\sqrt{\frac{c_2}{K_j}}, $$
> so that the approximation error produced in cycle $j$ is discounted by the factor $\mu^{N-1-j}$. Hence, early cycles are much less important for the final accuracy than late cycles. A fixed target-update frequency therefore, oversolves the early inner problems and wastes computation uniformly across all $N(\varepsilon)\sim \log(\varepsilon^{-1})$ outer iterations, which leads to the extra logarithmic factor. By contrast, the variable schedule balances this accumulated discounted approximation error using geometrically increasing $K_j$. Intuitively, one uses cheap/coarse Bellman approximations in early cycles and reserves accurate/expensive inner solves for the final cycles where they matter most. This is exactly the multilevel mechanism that removes the logarithmic overhead. We will add this explanation around Theorem 5.3.
>
> **Weakness 3:** Corrected, thanks!
>
> ## Questions
> **Question 1:** In principle, we believe the analysis can be extended to high-probability error bounds, but this would require replacing the current expectation-based control of the inner-loop error by a concentration result. In our tabular setting, this would likely amount to proving high-probability bounds for the stochastic inner iteration (e.g., SGD / stochastic approximation on a quadratic or strongly convex objective), and then allocating failure probabilities across the outer Bellman stages via a union bound. A simple conversion via Markov’s inequality is also possible, but would be quite loose and likely not reflect the true behavior of the algorithm. Obtaining a meaningful high-probability analogue would therefore require additional concentration arguments and is beyond the scope of the current paper.
>
> **Question 2:** The elimination of the $|S|\,|A|$ factor comes from our $(s,a)$-dependent inner-loop convergence in Proposition B.1. Lee \& He (see our references) use a standard SGD upper bound in their Proposition 2 on the expectation of the squared $L^2$-Norm, necessitating a technical bound of the expectation of the squared $L^2$-Norm of the Variance term (their Lemma 7). While our approach to first bound the expectation of the squared $(s,a)$-dependent distances (the $\Delta_k(s,a)$ at the top of page 14) requires a more careful treatment of applications of conditional expectations, it allows us to use a more straightforward bound on the now $(s,a)$-wise variance term (see lines 797-806) while the sum over the state action pairs from the $L^2$-Norm can be absorbed into the constants (see lines 833-836). We will add a short discussion to our revised manuscript when comparing the bounds.
>
> ## Additional Experiments
> We would like to point the reviewer to additional experiments on Atari-1: https://anonymous.4open.science/r/ICDQN-6F0C

---

> > ### Author Rebuttal · Reviewer_TWsn · 2026-03-31
> >
> > I thank the author for the detailed response and all my concerns are addressed.

---

### Official Review · Reviewer_NaYf · 2026-02-20

**Soundness:** 3
**Presentation:** 3
**Significance:** 3
**Originality:** 3
**Overall Recommendation:** 4
**Confidence:** 4

**Summary:**

This paper studies Q-learning with a target network, focusing on developing non-asymptotic guarantees and shedding light on the role of the target network update frequency. The authors show that the target update frequency (TUF) is indeed an important algorithmic design parameter and that, when chosen optimally, the convergence rate achieves an $\epsilon^{-2}(1-\gamma)^{-5}$ sample complexity.

**Compliance With Llm Reviewing Policy:**

Affirmed.

**Final Justification:**

I thank the authors for the detailed feedback.

(1) My concerns have been addressed.

(2) Thank the authors for the quantitative comparison; including it in the paper would help better position the results. It appears that the dependence on $1/(1-\gamma)$ can potentially be improved to $1/(1-\gamma)^4$ rather than $1/(1-\gamma)^5$. It may be worth mentioning this.

Overall, my concerns have been addressed, and I have increased my score to weak accept.

**Key Questions For Authors:**

See Strengths And Weaknesses.

**Limitations:**

Yes

**Strengths And Weaknesses:**

The paper is well written, and the results appear to be correct.

(1) The viewpoint of Q-learning with a target network as an approximate Bellman iteration—where the Bellman target is iteratively computed via SGD—is not new.

It is well understood in the literature that fitted Q-iteration can be viewed as approximate Bellman iteration in which the Bellman target is estimated via sample average approximation, while Q-learning with a target network can be viewed as approximate Bellman iteration in which the Bellman target is estimated via stochastic approximation (i.e., SGD). For example, [1] provides a non-asymptotic analysis of Q-learning with a target network under linear function approximation, which includes the tabular setting as a special case. This viewpoint is illustrated in detail in that paper, and the achieved sample complexity is comparable to the one obtained here, with the bias–variance tradeoff made explicit in the bounds as well.

[1] Chen, Z., Clarke, J. P., & Maguluri, S. T. (2023). Target network and truncation overcome the deadly triad in-learning. SIAM Journal on Mathematics of Data Science, 5(4), 1078-1101.

(2) There do not appear to be significant technical challenges in the analysis. The inner loop is a standard SGD procedure (in fact, a linear update) under Markovian noise, and the outer loop corresponds to approximate Bellman iteration. Moreover, the authors do not consider function approximation in their analysis, which further limits the technical scope.

---

> ### Author Rebuttal · Authors · 2026-03-31
>
> We thank the reviewer for reviewing our paper and the valuable feedback!
>
> We were not aware of the paper the reviewer is referring to. This is a very nice Q-learning article, with rather orthogonal contributions to our article. We will include a discussion of its results and setting in our revised manuscript.
>
> - **Article you are referring to**: The article manages to extend the known tabular Bellman/SGD point of view towards linear function approximation (with cyclic resetting of Q-approximations to $0$ and constant per-cycle learning rates). They achieve a very nice sample complexity with additional logarithmic terms.
>
> - **Our article**: The article targets the optimal choice of (increasing) inner cycle length (without resetting and per-cycle decreasing learning rates) to improve the $\epsilon$-dependency of the sample complexity. Additionally, a more careful $(s,a)$-wise SGD analysis improved the dependency of the sample complexity on $|\mathcal S|\,|\mathcal A|$ with respect to prior literature. It is quite appealing that $\epsilon^{-2}$-sample complexity (without log-terms!) can be achieved for Q-learning with the simple change of increasing TUF. As acknowledged by the other reviewers, our theoretical considerations have interesting consequences for the choice of TUF in DQN, as we show with two bridge examples. LunarLander was already included in the present version, and Atari-1 can be found in the additional document: https://anonymous.4open.science/r/ICDQN-6F0C
>
> Here are specific comments to the the points you raised:
>
> **Relation to the literature:**
>
> We agree that the interpretation of Q-learning with a target network as an instance of approximate Bellman iteration, where the Bellman target is computed via stochastic approximation, is not new. This viewpoint has been discussed in prior work, including Lee \& He, which we already cite in the paper. The reference mentioned by the reviewer provides a similar perspective and is closely related to that line of work.
>
> Our contribution is therefore not the introduction of this viewpoint, but rather the first analysis of how the TUF should be chosen in order to optimize the resulting sample complexity. In existing analyses (including Lee \& He and Chen et al.), the TUF is treated as a fixed parameter, and the resulting bounds contain an additional $\log(\epsilon^{-1})$ factor. In contrast, our analysis shows that by carefully designing the target update schedule, specifically through a geometrically increasing update interval, the logarithmic factor can be removed. Since the $\epsilon^{-2}$ dependence is known to be optimal, this result closes the remaining gap in the sample complexity analysis for this class of methods without modifying the $Q$-learning algorithm in such a way that it can not be applied in the function approximation setting. We do not agree with the reviewer that the analysis of Chen et al. (2023) yields a comparable sample complexity. Their error bound does not allow to derive optimal TUF design of our form. Furthermore, it should be noted that a key difference to our approach is the cyclic resetting of $Q$-approximations to $0$, which compared to our initialization scheme increases the initial error of each inner loop. This means that our tabular analysis is not a special case of the analysis of Chen et al., as the reviewer claims.
> Finally, in comparison to Lee \& He the dependence on the constant $|S|\,|A|$ is improved by choosing to take a more careful $(s,a)$-wise approach to the inner loop.
>
> **Technical challenges:**
>
> While the individual components of the algorithm are standard, the main technical challenge lies in analyzing their interaction under delayed target updates and deriving optimal TUF schedules.
>
> Conducting a refined $(s,a)$-dependent inner-loop convergence in Prop. B.1 allows us to eliminate one of the $|S|\,|A|$ factors compared to Lee \& He. They use a standard SGD upper bound in their Prop. 2 on the expectation of the squared $L^2$-Norm, necessitating a technical bound of the expectation of the squared $L^2$-Norm of the Variance term (their Lem. 7). Our approach to first bound the expectation of the squared $(s,a)$-dependent distances ($\Delta_k(s,a)$ at the top of page 14) requires a more careful treatment of applications of conditional expectations. This allows a refined $(s,a)$-wise variance term (see lines 797-806) while the sum over the state action pairs from the $L^2$-Norm can be absorbed into the constants (see lines 833-836). Moreover, removing the additional $\log(\epsilon^{-1})$ factor requires deriving our quasi-optimal TUF schedules. To show that these are quasi-optimal one has to handle non-uniqueness of the final error decomposition into contraction and approximation error.

---

> > ### Author Rebuttal · Reviewer_NaYf · 2026-03-31
> >
> > I thank the authors for their detailed response. However, my concerns are not fully addressed.
> >
> > (1) It might be better to downplay the cyclic viewpoint. Specifically, the sentence “This article provides a principled approximate dynamic programming view of target networks, interpreting periodic target fixing as a nested Bellman approximation scheme” in the first line of Section 6 is a bit misleading, given that this viewpoint already exists in the literature.
> >
> > (2) Can the authors provide a quantitative comparison of the sample complexity obtained in this work with those in (Lee & He) and (Chen et al.)? To enable a fair comparison, the authors can focus on the tabular setting (which is a special case of linear function approximation by setting the feature matrix to the identity). It would be interesting to see whether there are improvements in the dependence on $1/(1-\gamma)$ and $|\mathcal{S}||\mathcal{A}|$, given that the improvement in $\epsilon$ is only logarithmic.

---

> > > ### Author Response · Authors · 2026-04-01
> > >
> > > We thank the reviewer again for the feedback and for specifying their remaining concerns!
> > >
> > > ### (1)
> > >
> > > We agree that some of our formulations may be misinterpreted as claiming novelty of the cyclic viewpoint and will therefore carefully adapt all passes mentioning it with a special focus on the introduction and conclusion. In particular, we will refer to Chen et al. (2023) and Lee \& He (2020) when explaining the approximate Bellman setting. If the reviewer wishes, we could include a list of the exact adaptations we plan to make in another response, but due to space limit we will focus on answering the second concern.
> > >
> > > ### (2)
> > >
> > > **Our analysis:** $\mathcal{O}\Big(\frac{1}{\xi^2(1-\gamma)^5\epsilon^2}\Big)$, where $\xi$ is the smallest state-action probability.
> > >
> > > **Lee \& He (Thm. 3):** $\mathcal{O}\Big(\frac{|\mathcal{S}||\mathcal{A}|}{\epsilon^2(1-\gamma)^4\ln(\gamma^{-1})}\frac{L}{c^3}\ln\Big(\frac{1}{(1-\gamma)\epsilon}\Big)\Big)$, where $c$ and $L$ are the smallest and largest state-action probabilities (meaning $c=\xi$ and $L\ge1-\xi\frac{|\mathcal{S}||\mathcal{A}|-1}{|\mathcal{S}||\mathcal{A}|}$). To make the bounds directly comparable, we assume a uniform state-action sample distribution and set $\xi=c=L=\frac{1}{|\mathcal{S}|\ |\mathcal{A}|}$. We observe that our bound improves by a factor $\log(\epsilon^{-1}) |\mathcal S||\mathcal A|$ (as also mentioned in line 159-164 of our paper). Since $\log(\gamma^{-1})\sim (1-\gamma)^{-1}$ as $\gamma\to1$, the dependence in $\gamma$ improves in logarithmic terms of $(1-\gamma)^{-1}$. Note that the $|\mathcal S||\mathcal A|$ improvement is already visible in our fixed TUF setting (see Thm. 5.1 and Rem. 5.2), showing the advantage of the fine $(s,a)$-level considerations to a direct application of SGD results.
> > >
> > > **Chen et al. (derived from Thm 3.1, see below):** $\mathcal O\Big((t_\alpha +1) \frac{\log\Big(\frac{3}{\epsilon \sqrt{\lambda} (1-\gamma)^2}\Big) \log\Big(\frac{3e_0}{\epsilon}\Big) }{\lambda^3 (1-\gamma)^4 \log(\gamma^{-1})  } \epsilon^{-2} \Big)$, where $t_\alpha$ is the mixing time of the Markov chain $S_k$ (under behavior policy $\pi_b$) and $\lambda=\lambda_{\min}$ is the smallest eigenvalue of the feature covariance matrix $\Phi^T D \Phi$. In the tabular case as suggested by the reviewer this reduces to $\lambda = \min_{s,a} \mu(s)\pi_b(a; s)$, where $\mu$ is the stationary distribution of $S_k$ under $\pi_b$, corresponding to our $\xi$. In the setting of Chen et al, $t_\alpha$ is of order $\log(1/\alpha)$ and will be suppressed as constant here (improving the sample complexity). Similar to the comparison to Lee \& He, in a uniform state-action distribution setting our bounds improve by a factor $ \log(\epsilon^{-1})$ $|\mathcal S||\mathcal A|$ and even further logarithmic terms. Note that we don't want to overemphasize these additional logarithmic improvements here since the setting of Chen et al. is more general due to Markovian exploration assumptions and linear function approximation (which necessitates an additional truncation step). It would be very interesting to explore the extension of our optimal TUF design to this setting. This discussion will be added to our related work section.
> > >
> > > **Additional Derivation:** Theorem 3.1 in Chen et al. (2023) provides a clean non-asymptotic error decomposition allowing to derive the above sample complexity. This theorem states that
> > > $$\mathbb E[\lVert Q_T-Q^* \rVert_{\infty}] \leq \gamma^T \lVert Q_0-Q^* \rVert_{\infty} + \frac{2}{\sqrt{\lambda} (1-\gamma)^2}(1-\lambda \alpha)^{(K-t_\alpha-1)/2} + \frac{24\sqrt{\alpha (t_\alpha+1)}}{\lambda (1-\gamma)^2} =: E_1 + E_2 + E_3.$$
> > > To derive a sample complexity for $\mathbb{E}[\lVert Q_T-Q^* \rVert_{\infty}]\leq \epsilon$, we choose $T,K>0$ sufficiently large and $\alpha>0$ sufficiently small such that $E_1\le \epsilon/3$, $E_2\le \epsilon/3$ and $E_3\le \epsilon/3$. First, choosing $T\ge \log(3e_0/\epsilon) / \log(\gamma^{-1})$ with $e_0 = \lVert Q_0-Q^* \rVert_\infty$ yields $E_1\le \epsilon/3$. For bounding $E_3$ we require $\alpha \le \frac{\epsilon^2 \lambda^2 (1-\gamma)^4}{24^2\cdot 9 \cdot (t_\alpha+1)}$. Finally, we make use of this step size restriction to determine
> > > $$ K \ge t_\alpha + 1 + \frac{2}{\lambda \alpha}\log\Big(\frac{3}{\epsilon \sqrt{\lambda} (1-\gamma)^2}\Big) \ge t_\alpha + 1 + \frac{18 \cdot 24^2 \cdot\log\Big(\frac{3}{\epsilon \sqrt{\lambda} (1-\gamma)^2}\Big) (t_\alpha+1) }{\lambda^3 (1-\gamma)^4 \epsilon^{2} }.$$
> > > Hence, the sample complexity is of order
> > > $$T\cdot K \in \mathcal O\Big((t_\alpha +1) \frac{\log\Big(\frac{3}{\epsilon \sqrt{\lambda} (1-\gamma)^2}\Big) \log\Big(\frac{3e_0}{\epsilon}\Big) }{\lambda^3 (1-\gamma)^4 \log(\gamma^{-1})  } \epsilon^{-2} \Big). $$

---

### Official Review · Reviewer_vFtU · 2026-03-10

**Soundness:** 3
**Presentation:** 4
**Significance:** 3
**Originality:** 3
**Overall Recommendation:** 4
**Confidence:** 4

**Summary:**

This paper studies the impact of choosing the freezing time of the target network in Q-learning. The key innovation is that the authors assume the updated frequency of the target network is dynamic rather than being fixed. By applying a two-time scale analysis, the author shows that a previous logarithmic factor in the sample complexity bound (with respect to the accuracy $\epsilon$) can be avoided by geometrically increasing the update time of the target network. Experiments demonstrate the effectiveness of this dynamic update schedule.

**Compliance With Llm Reviewing Policy:**

Affirmed.

**Final Justification:**

The rebuttal adequately addressed my earlier questions/concerns. I find the overall analysis in the paper quite clean and insightful. However, my view on the logarithmic factor improvement remains. Thus, I wish to maintain my current score.

**Key Questions For Authors:**

1. In the grid world example, e.g., Fig. 2, even a large TUF (i.e., 10^5) still leads to some constant error. Is it because the paper updates the outer-loop by $Q_{n+1}=Q_n^{K_n}$? This transfers all noise from the previous inner loop to the next outer-loop, which may be too aggressive. Instead, if one uses a diminishing step-size to update the next $Q_{n+1}$, won’t Q-learning converge even with constant $K_n$? If such a diminishing step-size is used, will increasing TUF still have any benefit over constant $K_n$?

2. Will this technique of changing target-update frequency be able to improve the performance of double Q-learning?

**Limitations:**

Yes

**Strengths And Weaknesses:**

Strength:

1.The proposed topic is potentially impactful. Instead of fixing the target-network update frequency as a constant, the paper treats it as a configurable design choice that can be dynamically adjusted. The topic is relevant and paves the way for future hyperparameter optimization research.

2. This paper shows that dynamically choosing the target-network update frequency can lead to reduced sample-complexity: with a geometrically increasing target-network update time, the sample complexity bound can be improved by a logarithmic factor of the accuracy $\epsilon$.

3. The paper is well written and easy to follow.

Weakness:

1. Logarithmic improvement in sample complexity seems to be small (even though the numerical results seem to suggest more substantial gain).

2. The outer loop simply takes $Q_{n+1}=Q_n^{K_n}$, where $K_n$ is the freezing time of the target-network in round $n$. This may be too aggressive for constant $K_n$. Instead, if one uses a diminishing step-size to update the next $Q_{n+1}$, Q-learning may converge even with constant $K_n$. The paper neglects this possibility. It may then be unclear whether increasing $K_n$ is still better than constant $K_n$.

---

> ### Author Rebuttal · Authors · 2026-03-31
>
> We thank the reviewer for reviewing our paper and overall positive feedback. Below we provide detailed responses to the raised comments and questions.
>
> ## Weaknesses
>
> **Weakness 1:** Thank you for raising this point. While the improvement may appear modest at first glance, removing the logarithmic factor is in fact the strongest improvement possible in this setting. Even in the simplest setting of one state-action pair, where the Bellman operator becomes linear and $Q$-learning corresponds to estimating the expectation of the reward, $\varepsilon^{-2}$ attains the Cramer-Rao bound and is the optimal sample complexity. To the best of our knowledge, this is the first result in this setting that achieves the optimal rate $\varepsilon^{-2}$ without an additional $\log(\varepsilon^{-1})$ factor or modifying the $Q$-learning algorithm in a way that can not be applied to deep RL.
>
> Regarding the empirical gains, the observed improvements can be larger than the asymptotic logarithmic term might suggest because the analysis captures worst-case bounds, while the adaptive target update mechanism can reduce variance significantly in practice. We will clarify this point in the paper.
>
> **Weakness 2:** You are correct to state that diminishing step-sizes (satisfying Robbins-Monro conditions) will lead to convergence, even in the case of constant $K_n$. The key idea is similar. Diminishing step-sizes reduce the variance while slowing down the convergence of the exact Bellman iteration. In terms of sample complexity, even with diminishing step-sizes, one cannot improve the $\varepsilon^{-2}$-complexity. Thus, we focused on improving the existing bounds in the literature by using increasing TUFs, which gives the optimal rate of convergence. In the revision, we will add a theoretical comparison between increasing TUFs and diminishing step-sizes.
>
> ## Questions
> **Question 1:** The constant error in the examples stems from the fact that using the same initial step-size for all cycles means that inner loops with a fixed length cannot converge to accuracy more than a prescribed $\epsilon$ that depends on the initial step-size (see Thms 5.1 and 5.3). As discussed in the answer to W2, if we were to choose an initial step-size schedules for the cycles that diminishes over time, we can still retrieve convergence even with constant TUF.
>
> While we did not compare ICQL against Periodic Q-learning with diminishing step-size schedules, we have tested ICQL against Q-learning when both methods are using adaptive step-size schedules. More precisely, we tested both DQN and ICDQN with the Adam optimizer instead of SGD. The results show similar behavior as in the SGD case: https://anonymous.4open.science/r/ICDQN-6F0C
>
> **Question 2:** Good question! One of the reasons for overestimation in $Q$-learning is a poor estimation of $Q$-values which, combined with the maximum forces overestimation for the bootstrapping. Since the idea of increased TUF is to better estimate the optimality operator (and thus the $Q$-values) this can be seen as an approach that addresses overestimation as well. On small tabular examples we see exactly this effect. However, the main impact comes from the better estimation of the $Q$-values, regardless if originally over- or underestimation was present. Similarly, in double $Q$-learning both copies will be estimated better, so there is no reason not to combine both! Indeed, we incorporated our increasing target update schedule into a variant of BTR [1] which includes double-Q and is current SOTA in terms of walltime. This incorporation showed promising results on Atari-1, see link above.
>
> [1] T. Clark et al., Beyond the rainbow: High performance deep reinforcement learning on a desktop pc, ICML, 2025

---

> > ### Author Rebuttal · Reviewer_vFtU · 2026-04-03
> >
> > The response partially addresses my questions. Since you said you plan to provide theoretical comparison between increasing TUFs and diminishing step-sizes, will diminishing step-sizes lead to the same removal of logarithmic factor?

---

> > > ### Author Response · Authors · 2026-04-03
> > >
> > > We thank the reviewer again for the feedback and for specifying their remaining concerns!
> > >
> > > It is unclear if the removal of the logarithmic factor can be achieved with diminishing step-sizes and constant $K$ using some method other than the one presented in the paper, so we can not give a definitive answer to this question. In short, the key insight is: Previous literature does not contain results removing the logarithmic term with constant $K$ and diminishing step-sizes and using our approach strictly requires increasing $K$ even for diminishing step-sizes in order to remove the logarithmic term. What we plan to discuss in the revised manuscript is the following:
> > >
> > > **Comparison to previous literature:** To the best of our knowledge, there is not yet an analysis of the case $K>1$ that does not rely on similar techniques to the ones used in our paper. All of these analyses have the logarithmic factor, as they are using fixed $K$. See for example [3] for a study with constant outer-loop step-sizes but diminishing inner-loop step sizes, and [2] for results with constant step-sizes under linear function approximation. In the case of regular $Q$-learning, where $K=1$, there are different approaches to proving sample complexity bounds, for instance using the convergence result of Robbins-Monro. However, to the best of our knowledge, convergence results are either non-qualitative, meaning we can not derive sample-complexity results from them, or yield sample complexities with a logarithmic term. For the latter, we refer to recent references [1,4] and the references therein. (Note that in [4] the logarithmic factor is suppressed in the table at page 3).
> > >
> > > [1] Z. Chen et al., A Lyapunov Theory for Finite-Sample Guarantees of Markovian Stochastic Approximation, Operations Research 2024
> > >
> > > [2] Z. Chen et al., Target network and truncation overcome the deadly triad in-learning. SIAM Journal on Mathematics of Data Science 2023
> > >
> > > [3] D. Lee \& N. He, Periodic q-learning, Conference on Learning for Dynamics and Control, 2020
> > >
> > > [4] Q. Guannan \& A. Wierman, Finite-time analysis of asynchronous stochastic approximation and $ Q $-learning, Conference on learning theory, 2020
> > >
> > > **Diminishing step-sizes with our approach:**
> > > Diminishing step-sizes can be incorporated into our approach by rescaling $\alpha_n^{(k)}$ with a diminishing factor $1/n^p$ for some $p\ge 1/2$. From equation (10) in our paper it can then be deduced that our Theorem 4.5 still holds up to an additional constant factor (every $\xi$ is scaled with $\alpha_{n}^{(k)}\propto 1/n^{p}$, which cancels all factors $(\alpha_n^{(k)})^2\propto 1/n^{2p}$ from the reduced noise error term). Thus, diminishing stepsizes only change a constant in the upper bound of Thm. 4.1 on the inner loop. The convergence result of Corollary 4.6 and the analysis of the optimal fixed $K$ in Thm. 5.1 both use the splitting of the error from Equation (18). A change of constants in the inner loop bound does not affect the qualitative behavior of these terms, meaning that even with diminishing step-sizes the logarithmic term prevails as long as we only allow for fixed $K$.

---

### Official Review · Reviewer_MuF8 · 2026-03-12

**Soundness:** 2
**Presentation:** 2
**Significance:** 3
**Originality:** 3
**Overall Recommendation:** 4
**Confidence:** 4

**Summary:**

This paper provides a systematic theoretical analysis of the Target Update Frequency (TUF), a cornerstone stabilization mechanism in Deep Q-Learning that has long been treated as a heuristic "black box." The authors reformulate Q-learning with periodic target updates as a nested optimization scheme: an outer loop applying an inexact Bellman operator and an inner loop performing stochastic approximation.\
The paper delivers a finite-time convergence analysis under asynchronous Markovian sampling and identifies a fundamental bias-variance trade-off governed by the update period $K$. The authors prove that fixed update schedules are suboptimal and propose ATQL (Accuracy-Triggered Q-Learning), an adaptive strategy where the update interval evolves based on the progress of the inner loop.

**Compliance With Llm Reviewing Policy:**

Affirmed.

**Final Justification:**

Thank you for your response. I understand your setting and results much better now. I have raised the score to 4.

**Key Questions For Authors:**

Growth Rate of $K$: Your theory suggests $K$ should increase as training progresses. Does the optimal $K$ have an upper bound, or should it grow to infinity to ensure the final convergence of the outer Bellman iteration?\
Interaction with Double Q-Learning: Since Double Q-Learning also targets bias reduction, how does your frequency-control strategy interact with the decoupled architecture of Double Q? Is ATQL complementary to it?\
State Visitation Impact: How does the distribution of state visits (asynchronous vs. synchronous) affect the bias term in Theorem 1? Does a high-variance visitation pattern necessitate a significantly larger $K$?\
Practical Implementation: In high-dimensional spaces, the "accuracy trigger" $\epsilon_n$ might be hard to tune. Have you considered using a relative change in the value function instead of an absolute TD-error threshold?

**Limitations:**

Tabular and Linear Scope: The primary analysis is conducted in tabular and linear settings. While these offer clear theoretical insights into the bias-variance trade-off, the complex optimization landscapes of deep RL (non-convexity) might introduce additional dynamics (e.g., "dead neurons" or "rank collapse") that the current TUF theory does not fully account for.\
 The complexity of adaptive strategies: The proposed ATQL algorithm requires real-time monitoring of TD errors. The author should point out that in very large-scale parameter models, this kind of monitoring itself brings about computational load and its sensitivity to hyperparameters.

**Strengths And Weaknesses:**

Strengths\
Addressing a Fundamental Gap: While target networks are essential for the stability of algorithms like DQN, most theoretical literature focuses on step-size decay or overestimation bias. This work is among the first to rigorously quantify "how long we should fix the target" to balance stability and speed.\
Novel Theoretical Framework: By framing the process as an Approximate Power Iteration (API), the authors bridge the gap between classical Dynamic Programming theory and modern deep RL practices.\
Adaptive Strategy (ATQL): The transition from theory to a practical algorithm (ATQL) is well-executed. Moving away from static hyperparameters toward "accuracy-triggered" updates provides a principled way to manage training dynamics.\
Rigorous Convergence Bounds: Providing finite-time bounds under asynchronous sampling is a non-trivial contribution that makes the results applicable to more realistic RL scenarios.\
 Weaknesses\
Empirical Scale: The experimental validation is primarily conducted in tabular environments. While this aligns with the theoretical nature of the paper, ICML reviewers often look for at least one "bridge" experiment in a function approximation setting (e.g., a simple Neural Network on a classic control task) to verify if the qualitative behavior of $K$ matches the theory.\
Generalization to Non-linear Function Approximation: The theory is built on tabular/linear assumptions. In Deep RL, the interaction between the TUF and the non-convex landscape of neural networks might introduce complexities not captured here. A brief discussion on the limitations or extensions to the non-linear case would be valuable.\
Monitoring Overhead: The ATQL algorithm requires calculating a mean TD-error to trigger updates. In large-scale distributed systems, calculating this global metric might introduce non-negligible communication or synchronization overhead.

---

> ### Author Rebuttal · Authors · 2026-03-31
>
> We thank the reviewer for the positive feedback and constructive questions. Based on your suggestions, we have conducted a "bridge" experiment (Atari-1), see response to W1. This example shows that the theoretically suggested behavior is matched in practice. Considering your overall positive feedback, we hope that we can convince you to increase your final score.
>
> ## Weaknesses
> **W1 Empirical Scale:** Indeed, we did not do much experimental validation outside of tabular environments because of the theoretical scope of the paper. As a bridging example, we ran some simulations on Atari-1. We included DQN with SGD, as in the LunarLander example from the paper, DQN with ADAM to be closer to practice, and finally BTR [1], which is SOTA with regards to walltime. You can find these experiments here: https://anonymous.4open.science/r/ICDQN-6F0C
>
> [1] T. Clark et al., Beyond the rainbow: High performance deep reinforcement learning on a desktop pc, ICML, 2025
>
> **W2 Generalization:** We agree that extending the investigation of target update frequencies to a deep neural network setting is an important direction. Our goal in this work is to provide the first principled characterization of how the target update frequency should be chosen, and to do so in a setting where the effect of delayed targets can be isolated cleanly. In particular, our analysis identifies the bias–variance trade-off induced by freezing targets and establishes convergence guarantees without the additional $\log(\epsilon^{-1})$ factor that appears in most Q-learning theory papers.
>
> Extending the results in this article to deep RL would require combining our analysis with convergence guarantees for SGD/Adam in non-convex optimization, which remains a major open problem in modern deep learning theory itself. Since neural network training is generally only understood to converge to local stationary points with convergence rates that highly depend on the local geometry of the loss landscape, isolating the algorithmic effect of the target update frequency becomes substantially more difficult. We will clarify this scope in the conclusion section of our paper and highlight that the presented analysis should be viewed as a first theoretical step toward principled target-update schedules, which may guide the design of delayed target networks in deep RL.
>
> **W3 Monitoring Overhead:** Thank you for this insightful point. Our accuracy-triggered target update is based on the mean TD-error, which in the tabular setting corresponds to the Bellman residual and is proportional to the gradient of the squared TD-loss used in the inner-loop optimization. In this sense, the trigger can be interpreted as a residual-based stopping criterion for the inner Bellman iteration. Extending such stopping rules to large-scale deep RL systems would require more sophisticated criteria from non-convex optimization. One possible practical direction is the reviewer’s suggestion of monitoring relative changes in the value function, which could provide a scale-invariant stopping signal. We will add a discussion of this alternative in the revision.
>
> ## Questions
> **Q1 Growth rate:** Our results suggest that $K$ should increase with training to reduce variance while maintaining contraction of the (approximate) Bellman operator. To deduce convergence, we require $K$ to grow to infinity. However, for any target optimality accuracy $\varepsilon>0$ our theory provides a bound on $K(\varepsilon)$, it only grows to infinity as $\varepsilon$ goes to $0$.
>
> **Q2 Double-Q:** Good question! One of the reasons for overestimation is a poor estimation of $Q$-values which, combined with the maximum forces overestimation for the bootstrapping. Since the idea of increasing TUFs is to better estimate the optimality operator (and thus the $Q$-values) this can be seen as an approach that addresses overestimation as well. On small tabular examples we see exactly this effect. However, the main impact comes from the better estimation of the $Q$-values, regardless if originally over- or underestimation was present. Similarly, in double $Q$-learning both copies will be estimated better, so there is no reason not to combine it with ICQL or ATQL. Indeed, ICBTR from the additional Atari-1 experiment combines double-Q with increasing target update intervals.
>
> **Q3 State Visitation:**  The effect of a high-variance visitation pattern decreases the concentration property of the approximate Bellman operator. This effect can be seen from Proposition B.1, where the constant $\xi$ directly impacts the contracting term. Hence, the step size schedule naturally depends on $\xi$, and therefore, also the inner-loop convergence behavior, which results in a factor $1/\xi^2$ in the overall computational cost; see Equation (27). Consequently, a lower values of $\xi$ increases the target update intervals by a constant.
>
>
> **Q4 Practical Implementation:** See Weakness 3.

---

> > ### Author Rebuttal · Reviewer_MuF8 · 2026-04-03
> >
> > Thank you for your response. I understand your setting and results much better now.

---

### Decision · Program_Chairs · 2026-04-30

**Decision:**

Accept (regular)

**Comment:**

The paper provides a principled theoretical analysis of target update frequency (TUF) in Q-learning, reframing it as a design parameter rather than a heuristic. By modeling target updates as a nested optimization process, the authors derive finite-time convergence guarantees in the tabular setting. Their main result shows that constant TUF schedules are suboptimal, while geometrically increasing update intervals eliminate an avoidable logarithmic factor in sample complexity. The analysis is technically sound within its assumptions, and the resulting design insight, that target updates should become less frequent as learning progresses, is both intuitive and actionable.

Overall, the work is mathematically careful and contributes a clean, insightful framework, though its scope is limited to tabular/linear settings. The “optimality” claim should be interpreted relative to the paper’s derived bounds rather than in a universal sense, and the practical impact beyond theory remains somewhat uncertain. The authors should clarify the novelty relative to prior work, temper the optimality claims, and better connect the theory to practice, especially by discussing extensions to deep RL and comparisons with alternative mechanisms such as diminishing step sizes.